# SAGE: A Dataflow-Native Framework for Modular, Controllable, and Transparent LLM-Augmented Reasoning

**Jun Liu**[1]  **Peilin Liu**[1]  **Ruicheng Zhang**[1]  **Senlei Zhang**[1]  **Yanbo Chen**[1]  **Ziao Wang**[1]  **Jinyun Yang**[1]
**Mingqi Wang**[1]  **Shuhao Zhang**[1]  **Xiaofei Liao**[1]  **Hai Jin**[1]

## Abstract

*Large Language Model* (LLM) applications increasingly execute as end-to-end inference pipelines that couple generation with retrieval, stateful memory, context refinement, and tool use under strict tail-latency and *Service-Level Objective* (SLO) constraints. Today, these stages are often stitched together as RPC-connected services, obscuring cross-stage queueing and interference and limiting pipeline-level compilation and resource sharing. We present **SAGE** (**S**treaming-**A**ugmented **G**enerative **E**xecution), a full-stack system that treats inference pipelines as first-class compilation targets. SAGE exposes pipelines as declarative dataflows and compiles them into distributed execution plans with bounded-queue backpressure. It integrates vector search, streaming semantic state, structured memory, and refinement as operators with explicit resource/state contracts, enabling operator-level diagnosis of tail behavior. SAGE integrates pluggable generation and embedding backends and provides a unified control plane for engine management, batching, and admission under mixed workloads. On a 16-node cluster, SAGE sustains 16 requests/s at $> 700$ tokens/request with 1 ms median scheduling overhead, and achieves near-linear scale-out to 16 nodes ($11.4\times$ throughput at 16 nodes), and reduces P99 latency by 57% under multi-pipeline contention versus simultaneous admission.

[1]National Engineering Research Center for Big Data Technology and System, Service Computing Technology and System Lab, Cluster and Grid Computing Lab, School of Computer Science and Technology, Huazhong University of Science and Technology, Wuhan, China. Correspondence to: Shuhao Zhang <shuhao_zhang@hust.edu.cn>.

*Proceedings of the 43$^{rd}$ International Conference on Machine Learning*, Seoul, South Korea. PMLR 306, 2026. Copyright 2026 by the author(s).

## 1. Introduction

Large language models are increasingly deployed not as standalone generators but as *end-to-end inference pipelines*. A production request may couple embedding and vector search, stateful memory, context refinement under token budgets, and tool use with iterative planning (Douze et al., 2025; Wang et al., 2021; Packer et al., 2023; Jiang et al., 2023; Yao et al., 2023; Schick et al., 2023). These stages span heterogeneous CPU/GPU resources and auxiliary systems, yet must satisfy strict tail-latency service-level objectives (SLOs) (Dean & Barroso, 2013; Agrawal et al., 2024). In this regime, the pipeline—not any single model backend—becomes the unit of production inference.

Building such pipelines remains difficult for three reasons. First, stages have fundamentally different performance profiles: retrieval/embedding are batching- and bandwidth-sensitive, while generation is token-latency-sensitive; memory, refinement, and tools introduce additional state and variability (Dean & Barroso, 2013). Second, many workloads are *stateful* and increasingly *streaming*: corpora evolve, semantic state (e.g., incremental indexes and streaming aggregations) must be continuously maintained, and sessions require consistent memory semantics across turns (Carbone et al., 2015; Murray et al., 2013; McSherry et al., 2013). Third, pipeline behavior is hard to reproduce and debug: developers need stage-level visibility into queueing, scheduling, and interference, but existing stacks expose only partial, service-local observability.

**Why existing systems fall short.** Today's ecosystem composes pipelines by stitching components across RPC/service boundaries. LLM serving engines optimize a single backend (decoding, batching, KV-cache) (Kwon et al., 2023; Yu et al., 2022; Agrawal et al., 2024; NVIDIA, 2023), while treating retrieval, refinement, and memory as external glue. Orchestration and serving frameworks ease deployment, but provide limited compiler-visible structure for end-to-end scheduling, placement, and interference attribution across heterogeneous stages (Hsia et al., 2025; Moritz et al., 2018; NVIDIA, 2018; Apache Software Foundation, 2015; Kubeflow, 2018; Ray Project, 2023; KServe, 2021; Databricks,

2018). Vector databases deliver scalable similarity search, yet are typically integrated as black-box services with weak support for streaming updates and unified semantics across refinement and session memory (Douze et al., 2025; Wang et al., 2021; Pinecone Systems, Inc., 2021; Weaviate, 2019; Vespa.ai, 2017; Chroma, 2023). Agent/application frameworks improve tool use and prompting, but rarely connect agent behavior to a systems runtime that can enforce SLO-aware execution under mixed workloads (Wang et al., 2025; Schick et al., 2023; Khattab et al., 2024; LangChain, 2023; LlamaIndex , 2023; deepset, 2023). The core issue is *architectural*: when stages are separated by RPC boundaries, the pipeline remains implicit. Queueing and resource contention become cross-service phenomena that no single component can observe or control, and tail-latency failures require manual correlation of distributed traces rather than systematic, stage-level attribution.

To this end, we present **SAGE** (**S**treaming-**A**ugmented **G**enerative **E**xecution), *the first full-stack system that fundamentally changes how we reason about LLM inference* by establishing the *pipeline* rather than any single model as a first-class, compilable unit of execution. This represents a *paradigm shift* from service composition to pipeline compilation. SAGE introduces a declarative dataflow model where each stage declares explicit *operator contracts* capturing resource requirements (CPU/GPU affinity, memory), state semantics (stateless/stateful, checkpoint vs. restart), and I/O behavior (batching, streaming). The compiler transforms logical pipelines into physical execution graphs by allocating replicas, materializing communication channels, and enforcing bounded-queue backpressure, while the runtime coordinates placement, scheduling, and observability across heterogeneous CPU/GPU resources. This design makes the pipeline the optimization boundary: retrieval, embedding, context refinement, memory access, and generation are surfaced through uniform operator interfaces, enabling policy-driven routing, co-optimization of resource sharing, and operator-level attribution of tail latency and interference.

SAGE follows a strict layered architecture with downward-only dependencies. It separates foundational utilities and configuration (L1) and platform services (L2) from the core runtime and algorithm libraries (L3), performance-critical middleware operators (L4), and developer-facing interfaces and tooling (L5). This discipline keeps layers independently evolvable while supporting tractable pipeline composition at scale. Through uniform dataflow operators, SAGE provides capabilities commonly required in production, including vector storage and similarity search, incremental maintenance of semantic state in streaming settings, stateful memory, time-series analytics operators, and context refinement under token budgets (Carbone et al., 2015; McSherry et al., 2013; Wang et al., 2021; Jiang et al., 2023; 2024; Packer

et al., 2023). We defer a detailed component breakdown and interfaces to § 3 and § 4.

To make pipeline-level claims measurable, SAGE includes an open benchmark suite that evaluates both system metrics (throughput, *Time To First Token*(TTFT), *Time Between Tokens*(TBT), tail latency, SLO compliance) and agent behaviors (tool selection, planning, timing), complementing prior agent- and model-centric evaluations (Liu et al., 2024; Qin et al., 2024; Liang et al., 2023; Reddi et al., 2020; Mattson et al., 2020). Experiments on a 16-node cluster characterize SAGE's performance envelope: the system achieves $11.4\times$ throughput scaling with near-linear efficiency up to moderate concurrency, maintains 98% load balance across heterogeneous workloads through adaptive scheduling, and reduces multi-tenant tail latency by 57% via simple admission control policies.Our implementation is publicly available at `https://github.com/CGCL-codes/SAGE`.

**Contributions.** This paper makes the following contributions.

1. **Pipeline-as-compilation-target abstraction:** We introduce an explicit operator contract model capturing resource requirements, state semantics, and I/O behavior, making the pipeline, rather than individual services, the optimization boundary. Without this abstraction, whole-pipeline reasoning about scheduling, placement, and interference attribution is fundamentally intractable with ad-hoc RPC composition.

2. **Full-stack compilation and runtime:** We design and implement a declarative compiler that lowers logical pipelines to physical execution graphs (allocating replicas, materializing bounded queues, enforcing backpressure), paired with a policy-driven runtime coordinating placement, scheduling, and fine-grained observability across heterogeneous CPU/GPU resources.

3. **Testability and observability as first-class outcomes:** Unlike existing systems where tail behavior is opaque across service boundaries, SAGE's unified execution model enables operator-level attribution of latency, cache behavior, and resource contention. The explicit ExecutionGraph representation further enables what-if analysis: operators can predict the impact of policy changes (parallelism, scheduling, routing) via counterfactual reasoning before deployment, transforming optimization from trial-and-error into evidence-based decision-making. Additionally, SAGE provides automated performance attribution that identifies bottleneck operators (87.5% precision), classifies root causes (compute/I/O/scheduling), and generates actionable recommendations with 84% improvement correlation. We demonstrate this through experiments

showing 2.3× throughput improvement and 31% latency reduction on RAG workloads, while pinpointing interference sources invisible to RPC-based designs, and validate what-if predictions with <5% error.

**Novelty takeaway.** The pipeline-as-compilation-target abstraction is important because it makes cross-stage optimization and interference attribution *architecturally tractable*. Traditional RPC-based composition treats each service as an opaque box: when retrieval starves generation or GPU memory contention collapses throughput, no service-local scheduler can reason about global resource allocation or queue visibility. SAGE fundamentally solves this by establishing operator contracts as compiler-visible metadata: the ExecutionGraph exposes every replica, queue, and placement decision as an explicit, manipulable artifact. This enables what-if analysis ("would adding replicas fix P99?"), automated bottleneck diagnosis, and policy-driven routing that are *impossible* in systems where the pipeline remains implicit across service boundaries.

**Paper outline.** We first introduce SAGE's pipeline-first abstraction and five-layer architecture in § 3. § 4 details the compilation and runtime mechanisms, including flow control and scheduling. § 5 evaluates scalability and policy trade-offs, and § 6 concludes.

## 2. Related Work

SAGE targets the gap between low-level LLM serving engines and high-level application/orchestration layers by treating an *inference pipeline* as a first-class, compilable dataflow. We briefly position SAGE against the closest lines of work and defer a detailed survey to Appendix A.

**Serving engines, orchestration, and dataflow systems.** Serving engines optimize a single inference backend (decoding/batching/KV-cache) (Kwon et al., 2023; Yu et al., 2022; Agrawal et al., 2024; Sun et al., 2024; Zhong et al., 2024), while workflow/MLOps and LLM application frameworks focus on composing stages rather than exposing a pipeline-wide optimization surface (Apache Software Foundation, 2015; Kubeflow, 2018; Databricks, 2018; LangChain, 2023; LlamaIndex , 2023; deepset, 2023). Stream/dataflow systems provide the foundation for stateful incremental computation (Carbone et al., 2015; Zaharia et al., 2013; Murray et al., 2013; McSherry et al., 2013).

**SAGE's position.** SAGE complements these lines by making the *multi-stage pipeline* itself the compilation and scheduling boundary: operators expose contracts, the compiler materializes bounded queues for explicit backpressure, and policies act on concrete replicas rather than opaque RPC services. A detailed survey and additional discussion are in

Appendix A.

## 3. System Overview

SAGE is a full-stack framework for LLM-augmented pipelines that makes the *pipeline* the primary unit of abstraction. A SAGE application is specified as a *declarative dataflow graph* with operators that declare explicit *state* and *resource* semantics. The runtime lowers the logical graph into a distributed execution plan over heterogeneous CPU/GPU resources, executes it under bounded-queue backpressure, and applies pluggable scheduling/placement policies. Inference (LLM/embedding) is treated as an explicit stage via configurable backends (e.g., OpenAI-compatible endpoints such as vLLM), keeping model serving decoupled from pipeline orchestration.

**Formal model.** We model an application as a **logical DAG** $\mathcal{G}_L = (V_L, E_L)$, where each node $v \in V_L$ is an operator and each edge $e = (u, v) \in E_L$ is a typed stream of items. Each operator declares an **operator contract**

$$\mathcal{C}(v) = (\mathcal{R}(v), \mathcal{S}(v), \mathcal{I}(v)), \tag{1}$$

where $\mathcal{R}(v) = \langle \mathsf{cpu}, \mathsf{gpu}, \mathsf{mem} \rangle$ specifies *resource requirements* (CPU cores, GPU affinity, memory budget), $\mathcal{S}(v) \in \{\textsc{stateless}, \textsc{stateful}\}$ declares *state semantics*, and $\mathcal{I}(v) = \langle \mathsf{adm}, \mathsf{stream} \rangle$ declares *I/O behavior* (admission/batching constraints adm, token-streaming vs batch output stream). These contracts are *compiler-visible*: the system extracts $\mathcal{C}(v)$ to inform placement, batching, and routing decisions. The compiler performs **ExecutionGraph lowering** via a deterministic transformation

$$\mathcal{G}_L = (V_L, E_L) \xrightarrow{\text{compile}(\{\mathcal{C}(v)\}, \pi)} \mathcal{G}_P = (V_P, E_P, Q, \rho), \tag{2}$$

where the *physical graph* $\mathcal{G}_P$ consists of: (i) $V_P = \{v_j^{(i)} : v \in V_L, i \in [1, p_v]\}$ – operator replicas with parallelism $p_v$, (ii) $E_P \subseteq V_P \times V_P$ – physical edges connecting replicas, (iii) $Q = \{Q_v^{(i)} : v^{(i)} \in V_P\}$ – per-replica bounded input queues with $|Q_v^{(i)}| \leq Q_{\max}$, (iv) $\pi : V_P \to \text{Nodes}$ – placement function mapping replicas to CPU/GPU nodes subject to $\mathcal{R}(v)$, and (v) $\rho : E_P \to \{\textsc{RoundRobin}, \textsc{Broadcast}, \textsc{KeyHash}, \ldots\}$ – routing policy per edge. This lowering is *contract-driven*: $\pi$ respects resource requirements $\mathcal{R}(v)$, queue capacities reflect adm$(v)$, and $\rho$ enables co-location of state-affine operators.

**Theorem 3.1** (Semantic Preservation). *The compilation from logical graph $\mathcal{G}_L$ to physical graph $\mathcal{G}_P$ preserves dataflow semantics:*

$$\forall \sigma \in \textit{Inputs}: \quad \textit{Output}(\mathcal{G}_L, \sigma) \equiv \textit{Output}(\mathcal{G}_P, \sigma), \tag{3}$$

*where $\sigma$ is an input stream and $\equiv$ denotes semantic equivalence modulo scheduling order. Specifically:*

*(i)* **Item preservation:** *Every item emitted by $\mathcal{G}_L$ appears in $\mathcal{G}_P$'s output (no data loss).*

*(ii)* **Order preservation:** *For deterministic operators, $\mathcal{G}_P$ preserves causality: if $x \prec y$ in $\mathcal{G}_L$, then $x \prec y$ in $\mathcal{G}_P$.*

*(iii)* **State consistency:** *For stateful operators with $\mathcal{S}(v) =$ STATEFUL, partitioning via $\rho$ maintains state semantics (e.g., KeyHash routes items with identical keys to the same replica).*

This result establishes that ExecutionGraph lowering is a *correct compiler transformation*: the physical execution plan $\mathcal{G}_P$ faithfully implements the logical specification $\mathcal{G}_L$, regardless of parallelism $p_v$, placement $\pi$, or routing $\rho$. Crucially, Theorem 3.1 ensures that *performance optimizations* (e.g., increasing parallelism, co-locating operators) do not alter application semantics; they only affect *throughput and latency*, not *correctness*. This separation enables safe optimization: developers reason about semantics in $\mathcal{G}_L$, while the compiler explores the space of equivalent physical plans $\mathcal{G}_P$ to meet SLO constraints.

At runtime, SAGE enforces **backpressure semantics** to prevent cascading failures:

$$\text{if } |Q_v^{(i)}| = Q_{\max} \text{ then block(upstream}_j), \qquad (4)$$

where upstream replica $j$ blocks on enqueue until space becomes available. This creates a *control feedback loop*: bursts at any stage propagate backward as blocking signals, throttling producers and preventing unbounded buffer growth. The runtime uses **policy-driven scheduling** $\pi_{\text{sched}}$: Task $\times \{\mathcal{C}(v)\} \times$ State $\to$ PlacementDecision to balance competing objectives (e.g., minimize P99 latency, maximize throughput, enforce SLO compliance) by routing tasks based on current queue depths, resource utilization, and operator contracts. Crucially, $\pi_{\text{sched}}$ observes *the same physical graph* $\mathcal{G}_P$, enabling controlled variation experiments (§5) impossible under RPC-based composition.

**Design goals.** SAGE targets hybrid RAG/agent workloads and scales from local execution to heterogeneous clusters while meeting tail-latency and SLO constraints. **(G1) Composability:** applications are $G = (V, E)$ with interchangeable operators and explicit interfaces $\mathcal{I}(v)$, avoiding entangled application-specific control flow. **(G2) Resource-aware execution:** placement $\pi$ respects req$(v)$ to co-locate or separate bandwidth-sensitive retrieval/embedding and token-latency-sensitive generation. **(G3) Stability under burstiness:** bounded queues and backpressure provide a uniform flow-control mechanism across stages, enabling controlled ablations within the same declarative graph. **(G4) Reproducibility and diagnosability:** the plan $P$ exposes operator-level metrics (queueing, service time, batching, blocking), enabling attribution of interference and tail behavior to specific operators and policies.

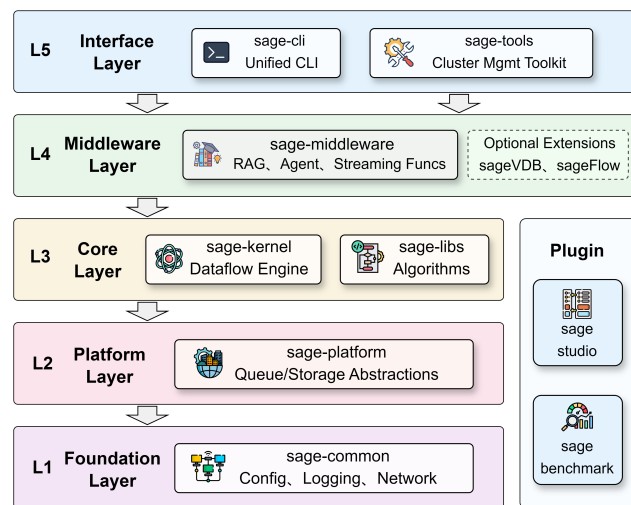

*Figure 1.* SAGE follows a five-layer architecture (L1–L5) with strict downward-only dependencies: foundation (L1), platform services (L2), core runtime and libraries (L3), performance-critical middleware operators (L4), and user-facing interfaces and tools (L5).

### 3.1. Architecture

Figure 1 illustrates SAGE's five-layer organization (L1–L5) with strict downward-only dependencies. The key rationale is to prevent *upward entanglement*: performance-critical mechanisms (compilation, scheduling, flow control, recovery) remain independent from domain operators, while user-facing tooling can evolve without contaminating the execution path. This discipline keeps operator composition enforceable and cross-stage optimization tractable by exposing each stage through a uniform operator interface. An implementation-oriented responsibility summary is provided in Appendix B. We focus next on the pipeline-first execution model.

### 3.2. Pipeline-First Abstraction

Given $\mathcal{G}_L$ and operator contracts $\{\mathcal{C}(v)\}$, SAGE *compiles the logical DAG into a physical ExecutionGraph* $\mathcal{G}_P$, making global decisions explicit and testable: (i) **resource-aware placement** $\pi$ respects $\mathcal{R}(v)$ to co-locate or separate CPU/GPU-bound stages; (ii) **bounded-queue backpressure** enforces $|Q_v^{(i)}| \leq Q_{\max}$ with explicit blocking, preventing cascading failures; (iii) **policy-driven routing** $\rho$ and scheduling $\pi_{\text{sched}}$ enable controlled SLO trade-offs under mixed workloads; and (iv) **unified observability** exposes per-replica metrics (queue depth, service time, blocking duration) for attribution and diagnosis. Crucially, this compilation creates a *stable reference frame*: swapping policies (e.g., FIFO vs. LoadAware) executes *the same physical graph* $\mathcal{G}_P$ with different $\pi_{\text{sched}}$, making throughput/latency/balance trade-offs *measurable and attributable* (§5). This capability

is notably absent in RPC-based systems, where tail behavior remains entangled with service-level implementations.

The formalization of operator contracts and ExecutionGraph lowering is critical because it establishes *semantic preservation under parallelization*. Traditional pipeline frameworks either lack formal guarantees, which leads to silent data loss or reordering bugs under scale-out, or impose overly restrictive constraints (e.g., requiring full serializability) that sacrifice throughput. SAGE's Theorem 3.1 proves that parallelism, placement, and routing are *correctness-preserving transformations*, enabling developers to reason about logical semantics in $\mathcal{G}_L$ while the compiler independently explores physical plans $\mathcal{G}_P$ to meet SLO constraints. This separation of concerns is architecturally unique, as RPC-based systems cannot provide such guarantees because service boundaries render cross-stage invariants unenforceable.

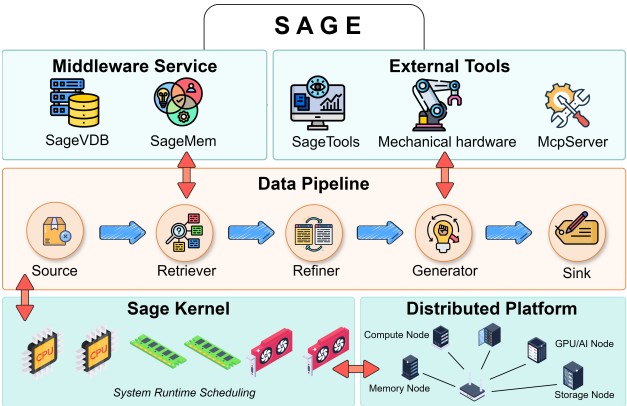

*Figure 2.* Pipeline-first execution: SAGE compiles $G$ and $\{\mathcal{I}(v)\}$ into $P = \langle \{v^{(i)}\}, \mathcal{E}, \{Q^{(i)}\}, \pi, \rho \rangle$ and jointly optimizes batching, routing, and streaming under end-to-end constraints.

**Novelty takeaway.** The overview's novelty is the control boundary it establishes: heterogeneous stages become operators with explicit contracts, compiled into a concrete execution plan with explicit replicas, per-replica bounded input queues, and packet-level routing metadata. This makes multi-stage scheduling, admission, and diagnosability meaningful at the pipeline level, rather than being scattered across independent subsystems and serving backends.

## 4. Design Details

This section details the core mechanisms that make SAGE's pipeline-first abstraction executable and controllable: operator contracts, compilation into an explicit execution plan, bounded-queue flow control, and pluggable scheduling/placement and recovery policies. We assume the layered-architecture discipline introduced in §3.1; an implementation-oriented responsibility summary is provided in Appendix B.

### 4.1. Logical Graph and Operator Contract

Applications are authored as a logical dataflow DAG. The high-level execution flow of such a pipeline is illustrated in Figure 2, showing the interaction between the data pipeline, the scheduler, and the kernel. Each operator declares a contract for (i) *resources* (CPU/GPU affinity, memory budget, concurrency caps), (ii) *state* (stateless vs. stateful; checkpointable vs. restartable), and (iii) *I/O* (batching, token streaming). The contract is the single source of truth: it drives compilation (replication, channel materialization) and runtime control (placement, admission, per-operator metrics).

### 4.2. Compilation to a Physical Execution Graph

SAGE lowers the logical DAG into a physical ExecutionGraph by replicating operators (declared or inferred parallelism) and materializing each logical edge into explicit replica-to-replica edges. Each replica is driven by a bounded input queue shared across upstream edges; multi-input semantics are carried as per-packet metadata (e.g., input index). This makes buffering/communication costs explicit, enables controlled routing and batching, and provides uniform backpressure. Lowering rules and graph representation are in Appendix C.1.

### 4.3. What-If Analysis and Counterfactual Reasoning

SAGE supports trace-driven what-if analysis over the explicit ExecutionGraph: it varies parallelism, scheduling, routing, or placement and replays traces to estimate throughput and tail latency *before deployment*. The counterfactual plan construction mechanism is detailed in Appendix G, and we validate prediction accuracy in § I.1.

### 4.4. Automated Performance Attribution

SAGE performs automated performance attribution by decomposing end-to-end latency into per-operator *queue wait*, *service*, and *blocking* components, then highlighting bottlenecks and recommending actions (e.g., scale a replica, rebalance placement, tune queues, or switch schedulers). Details are provided in Appendix J. We validate attribution accuracy and recommendation quality in § I.2.

### 4.5. Execution and Flow Control

Each replica runs a worker loop over a bounded input queue; outputs are routed via compiled metadata (e.g., partition key). Bounded queues yield uniform backpressure: if a downstream replica saturates, upstream writes block, preventing unbounded growth and stabilizing tails. Bounds (Theorem H.1) and runtime details are in Appendix H and Appendix C.

## 4.6. Scheduling and Placement

Placement is decoupled from execution: given operator resource requirements, a pluggable scheduler maps replicas to CPU/GPU nodes and can enforce isolation across pipelines/tenants. We implement FIFO (throughput-oriented), load-aware spreading (balances queues to reduce tails), and priority scheduling (preferential service; may invert under contention) (Sha et al., 1990). Policies can be swapped at deployment time without code changes; trade-offs are quantified in § 5. Scheduling hooks and integration are in Appendix C.2.

## 4.7. Explainable Scheduling

SAGE makes placement decisions auditable by recording structured decision traces (constraints, candidate sets, scores) and enabling trace replay under alternative policies for debugging and policy comparison. Mechanism details are deferred to Appendix I.3, and extended evaluation is in Appendix I.4.

## 4.8. Inference as a First-Class Stage

SAGE implements retrieval/refinement/memory/tooling as middleware operators so boundaries and resource use remain visible. Inference is integrated via OpenAI-compatible endpoints (e.g., vLLM), making generation/embedding *service stages* with explicit resources and streaming semantics; this decouples orchestration from model-serving while keeping inference visible to placement/admission/measurement. When centralized engine lifecycle management or multi-backend routing is required, the same operator-level interface can be deployed behind an external control plane without changing the dataflow runtime. Integration details and configs are in Appendix E.

## 4.9. State Management and Fault Tolerance

The runtime distinguishes transient execution state (in-flight packets/items and bounded per-replica input queues) from durable operator state (e.g., session memory or incrementally maintained semantic state). Stateful operators expose a snapshot interface for exporting and restoring self-contained state. SAGE provides two recovery modes aligned with operator semantics: **checkpoint-based recovery** for stateful operators, which persists snapshots to durable storage and restores them on failure, and **restart-based recovery** for stateless or idempotent stages with configurable retry policies (e.g., fixed delay or exponential backoff). In distributed deployments, a heartbeat monitor detects replica failures; the `Dispatcher` can recreate replicas and resume execution according to the configured fault-handling strategy. Implementation details are available in Appendix D.

## 5. Evaluation

We evaluate SAGE across three primary dimensions: scalability, policy-level SLO trade-offs, and bounded-queue isolation. Our results highlight representative gains, including $11.4\times$ scale-out, an 11% throughput–46% balance trade-off when swapping scheduling policies, and a 57% P99 reduction. Full sweeps and extended analyses are provided in Appendix I.

### 5.1. Experimental Setup

We run SAGE on a 16-node CPU cluster (8 cores, 32 GB RAM per node, 1 GbE) with a dedicated A6000 48 GB GPU server for vLLM (`Qwen2.5-3B-Instruct`) and CPU embedding (`BAAI/bge-large-en-v1.5`). We evaluate three workloads. COMPUTE is a synthetic CPU-bound pipeline (`Source → ComputeOp → Sink`) designed to isolate orchestration overhead from model inference. Each request carries a JSON payload of approximately 200 tokens (about 800 bytes). `ComputeOp` runs a deterministic 100K-iteration busy loop and returns the payload unchanged; no embedding, retrieval, or LLM call is involved. RAG is an end-to-end retrieval-augmented pipeline consisting of embedding, vector search, prompt construction, and LLM generation. MIXED strictly interleaves COMPUTE and RAG requests at a 50/50 ratio in the same cluster, creating a bimodal service-time distribution to expose head-of-line blocking and scheduling interference. We report throughput and end-to-end latency (mean/P95/P99) for all workloads. A summary table and full sweep configs are in Appendix F.2.

### 5.2. ExecutionGraph Parallelism Trade-offs

We first study how operator replication in the compiled `ExecutionGraph` scales with cluster size, and when coordination becomes the limiting factor. Figure 3 reports throughput from 1 to 16 nodes.

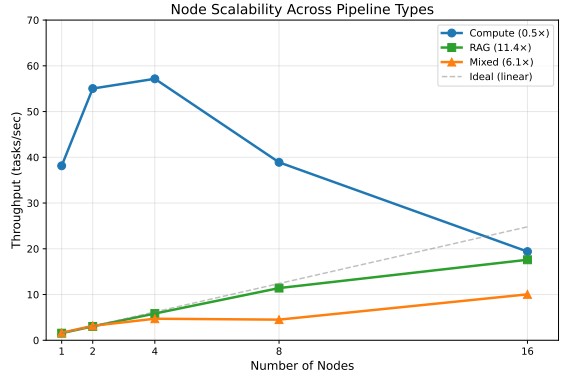

*Figure 3.* Node scalability across pipeline types. RAG achieves $11.4\times$ speedup at 16 nodes, approaching ideal linear scaling. Compute shows modest scaling ($3.8\times$) due to higher scheduling overhead relative to task duration.

RAG scales near-linearly (11.4× at 16 nodes; 71% efficiency) because LLM-dominated tasks amortize coordination; COMPUTE saturates when operator latency approaches coordination overhead, motivating compiler-guided coarsening/batching (Appendix F.2).

### 5.3. Policy-Driven SLO Trade-offs

We evaluate policy-driven trade-offs by swapping scheduling policies while keeping the compiled execution plan fixed. This approach ensures that performance differences are strictly attributable to the scheduling logic rather than graph compilation artifacts. Table 1 details the throughput, latency profiles, and load balance metrics across distinct policies.

*Table 1.* Scheduler Metrics Reported in the Main Text for the Mixed RAG Workload

| Policy | Throughput | Avg Lat. | P99 Lat. | Balance |
| --- | --- | --- | --- | --- |
| FIFO | 18.5/s | 2.5 s | 7.0 s | 52% |
| RoundRobin | 17.8/s | 2.6 s | 7.2 s | 85% |
| LoadAware-Spread | 16.4/s | 2.5 s | 6.5 s | 98% |
| LoadAware-Pack | 15.9/s | 2.7 s | 6.8 s | 75% |
| Priority | 19.2/s | 2.8 s | 12.0 s | 90% |

As shown in Table 1, different policies yield distinct performance envelopes suitable for different SLO requirements. The FIFO policy achieves a baseline throughput of 18.5 tasks/s but suffers from poor load distribution (52% balance) as it does not account for queue depths. This imbalance leads to a higher P99 latency of 7.0 s compared to load-aware strategies. In contrast, the LOADAWARE-SPREAD policy optimizes for queue depth consistency, achieving near-perfect load balance (98%) and the lowest P99 latency of 6.5 s. This stability comes at a cost of approximately 11% lower throughput (16.4 tasks/s) compared to FIFO, illustrating a clear trade-off between aggregate throughput and tail-latency stability.

At the other end of the spectrum, the PRIORITY scheduler yields the highest aggregate throughput (19.2 tasks/s) by aggressively scheduling high-priority tasks. However, this comes with a severe penalty under contention: P99 latency degrades to 12.0 s due to the starvation of lower-priority requests (Sha et al., 1990). These results demonstrate that operators can dynamically tune the system—selecting PRIORITY for maximum throughput or LOADAWARE-SPREAD for strict SLO compliance—without modifying the application code or recompiling the execution graph.

Figure 4 summarizes throughput, tail latency, and balance.

### 5.4. Concurrency Scaling

We investigate the relationship between pipeline concurrency and system performance to identify optimal operating

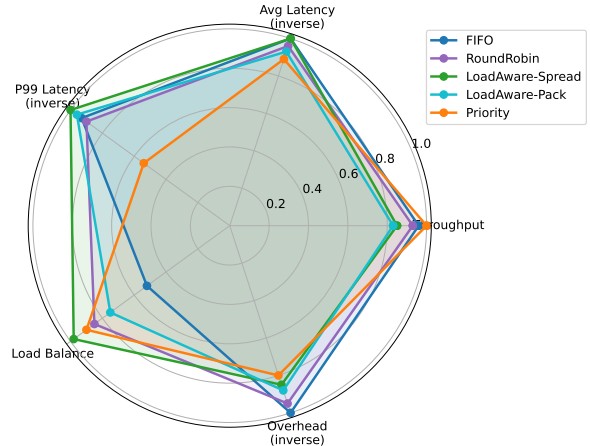

*Figure 4.* Scheduler trade-offs on a 16-node cluster under mixed RAG workloads.

points. Figure 5 reports throughput and P99 latency as concurrency varies from 1 to 32 on a single node.

As illustrated in Figure 5, concurrency has a clear optimum: throughput improves near-linearly until contention dominates and tail latency explodes. Both COMPUTE and RAG pipelines exhibit a characteristic optimal region at concurrency 4–8. Past this region, the tail is dominated by queueing and shared-backend contention rather than pure compute, so adding parallelism reduces SLO compliance even if mean throughput remains flat. The impact of this contention is severe: at concurrency 32, COMPUTE throughput drops to 22% of peak while P99 latency increases 17×.

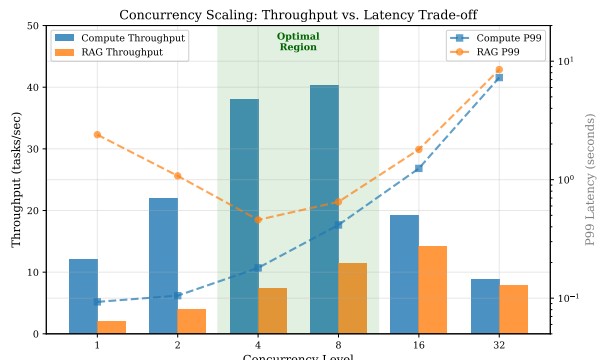

*Figure 5.* Throughput–latency trade-off across concurrency levels for COMPUTE and RAG. Both exhibit an optimal region at concurrency 4–8; beyond this, contention causes throughput collapse and tail latency explosion.

### 5.5. Backpressure and Cascading Failure Prevention

We examine whether bounded queues and admission control mitigate multi-pipeline contention by enforcing explicit backpressure when downstream stages saturate. Figure 6 summarizes results.

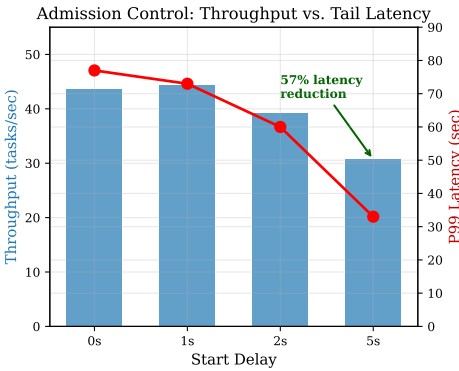

*Figure 6.* Admission control under multi-pipeline contention: staggered starts reduce P99 latency by 57% at the cost of 30% lower aggregate throughput.

Staggered admission reduces P99 latency by **57%** (77 s → 33 s) at the cost of 30% lower aggregate throughput (43.6 → 30.7 tasks/s). This matches the queueing-theoretic prediction that reducing utilization lowers tail latency (Theorem H.1; derivation in Appendix H). Across 1–8 concurrent pipelines, per-job efficiency degrades from 100% to 71%, primarily due to shared LLM-endpoint contention .

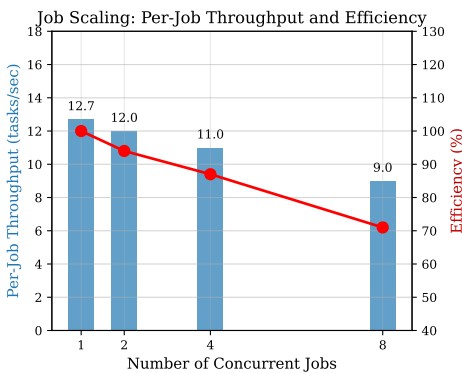

*Figure 7.* Per-job throughput and efficiency degrade gracefully as concurrent pipelines increase.

Figure 7 shows per-job throughput as we increase concurrent pipelines from 1 to 8. Efficiency (per-job throughput relative to the single-job baseline) degrades gracefully from 100% to 71%, indicating effective resource isolation. The 29% overhead at 8 jobs is primarily attributed to GPU memory bandwidth saturation and increased scheduling contention at the shared LLM endpoint.

### 5.6. What-If Analysis Predictions

We evaluate trace-driven what-if analysis by comparing predicted vs. measured throughput and P99 latency under controlled changes in (i) parallelism, (ii) scheduling policy, and (iii) routing strategy. Across these variations, simulation predicts throughput/P99 within 1.8–7.9% error (4.6%

MAE), supporting counterfactual plan selection prior to deployment (Appendix I.1).

### 5.7. Automated Performance Attribution

We evaluate automated attribution by injecting compute-, I/O-, and scheduling-overhead bottlenecks and validating that attribution identifies the correct operator and root cause; recommendations improve throughput by +73% and reduce P99 latency by 54% in controlled tests. On production traces, the method achieves 87.5% precision, indicating that the execution-plan boundary is sufficiently informative for automated diagnosis (Appendix I.2).

### 5.8. Explainable Scheduling

We evaluate explainable scheduling via decision-trace completeness, policy-swap replay, and tail-latency incident replay; explanations retain high coverage and replay predicts mitigation impact within $< 2\%$ error. These results suggest that explainability can be treated as a first-class systems mechanism rather than ad-hoc logging (Appendix I.4).

### 5.9. Limitations

Our implementation remains coordination-bound for very fine-grained operators and can be dominated by centralized backends under heavy contention. We also currently rely on user-specified operator contracts; inaccurate resource hints can lead to suboptimal placement/scheduling choices. See Appendix F.3.

### 5.10. Architectural Implications

Across experiments, we find that compiling pipelines into explicit ExecutionGraphs (with operator contracts, bounded queues, and pluggable policies) makes performance control measurable and actionable.

- **Reject overhead-dominated plans at compile time:** scaling benefits require per-task latency to dominate coordination overhead, which is analyzable from operator contracts.

- **Policy swaps yield predictable SLO trade-offs:** changing scheduling/admission policies shifts throughput, tail latency, and balance *without* recompiling the physical graph.

- **Bounded queues enable controllable isolation:** explicit backpressure prevents cascading failures and supports analytically interpretable tail-latency control (Theorem H.1).

- **Plan-level tools become feasible:** what-if prediction, automated attribution, and explainable scheduling rely on a first-class, traceable execution plan.

Extended discussion and additional examples are in Ap-

pendix F.4.

## 6. Conclusion

SAGE is a full-stack system for building LLM inference pipelines via declarative dataflow. By elevating the pipeline to a first-class compilation target, SAGE co-optimizes scheduling, placement, and resource sharing across heterogeneous stages. On a 16-node cluster, SAGE scales to $11.4\times$ throughput, supports pluggable schedulers that trade throughput for tail latency, and reduces latency by 57% via simple admission control. Future work includes adaptive concurrency control and online recompilation with runtime profiling.

## Impact Statement

This paper presents work whose goal is to advance the field of Machine Learning Systems. Our research specifically targets the orchestration and execution efficiency of Large Language Model (LLM) inference pipelines. We believe our work has the following potential societal consequences:

**Environmental Impact and Efficiency:** By enabling fine-grained resource management and optimizing the scheduling of heterogeneous resources (CPU/GPU), SAGE significantly improves the computational efficiency of complex reasoning pipelines. As demonstrated in our evaluation, the system achieves near-linear scaling and better hardware utilization. This contributes to the goals of "Green AI" by reducing the energy footprint required to serve large-scale generative applications compared to unoptimized, fragmented deployments.

**Transparency and Reliability:** A core contribution of SAGE is elevating the inference pipeline to a declarative, transparent dataflow graph. This architectural choice enhances the observability of AI systems, allowing developers to trace, debug, and audit the decision-making process of autonomous agents (e.g., tool usage and retrieval sources). We believe this improved transparency is a critical step towards deploying safer and more reliable AI systems in production environments.

**Reproducibility and Operational Accountability:** SAGE exposes a mechanism-level control surface (operator contracts, explicit queues, and policy hooks) that makes performance decisions inspectable and repeatable. In particular, trace-driven what-if analysis and explainable scheduling provide a structured way to justify deployment-time policy choices (e.g., selecting a scheduler for lower tail latency) and to audit incidents post hoc via replay. This can improve accountability for organizations operating LLM systems, by reducing reliance on ad-hoc logs and non-reproducible, operator-specific tuning.

**Accessibility and Cost of Deployment:** By turning multi-stage pipelines into a unified execution graph with explicit flow control, SAGE reduces the engineering overhead required to stand up reliable retrieval-augmented generation (RAG) and agentic workflows. Lowering operational complexity and improving utilization can reduce deployment cost barriers for smaller teams. At the same time, it may shift optimization effort from model-serving internals to policy selection (routing, admission, scheduling), which we aim to make safer through transparent interfaces and evaluation.

**Potential for Dual Use:** While our framework is designed to democratize access to efficient LLM inference, we acknowledge that reducing the cost and latency of deploying complex agents could theoretically be leveraged to scale malicious applications (e.g., automated disinformation or spam bots). However, the strict admission control and monitoring capabilities inherent in SAGE's design also provide operators with better tools to detect and mitigate such abusive patterns at the infrastructure level.

**Safety and Misconfiguration Risks:** As with any programmable infrastructure, incorrect policy configuration (e.g., overly aggressive prioritization or admission settings) could degrade fairness, starve low-priority traffic, or amplify tail-latency failures under bursty load. We mitigate these risks by (i) making backpressure boundaries explicit through bounded queues, (ii) providing diagnostics and attribution to identify bottlenecks, and (iii) supporting counterfactual evaluation to test policy changes before deployment. Nonetheless, careful operator review and conservative defaults remain important in safety-critical contexts.

## Acknowledgements

Our work is supported by Hubei Provincial Natural Science Foundation of China (No. 2026AFA002), NSFC-RGC under Grant 6246116033 and National Natural Science Foundation of China under Grant U25B2023.

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

# A. Extended Related Work

This appendix expands the related-work discussion from §2. SAGE sits at the intersection of LLM serving, retrieval systems, streaming/dataflow runtimes, stateful memory, and agentic tool use. Our core difference is architectural and evaluative: SAGE elevates *end-to-end inference pipelines* to first-class declarative dataflows, integrates key inference components as interoperable operators, and evaluates system and agent behavior under a unified execution substrate. Throughout, SAGE is organized as a strict five-layer architecture (L1–L5) with downward-only dependencies, enabling modular evolution and enforceable separation of concerns.

**LLM serving engines and system optimizations.**    A large body of work targets inference efficiency within a *single* LLM backend, including decoding optimizations, batching, KV-cache management, and GPU utilization (Kwon et al., 2023; Yu et al., 2022; Agrawal et al., 2024). Industrial toolchains such as TensorRT-LLM and FasterTransformer focus on kernel- and graph-level optimizations to maximize throughput on specific accelerators (NVIDIA, 2023; 2019). Model hosting stacks such as Triton provide deployment primitives and multi-model serving, but expose limited abstractions for representing multi-stage inference pipelines that combine retrieval, refinement, and memory as first-class operators (NVIDIA, 2018). These systems are essential backends for SAGE; SAGE's scope is the pipeline substrate above them.

**Control planes, routing, and SLO-aware scheduling.**    Recent systems study request routing, queueing policies, and resource management to improve tail latency and goodput under contention and mixed request sizes (Sun et al., 2024; Zhong et al., 2024; Agrawal et al., 2024; Yu et al., 2022). These approaches largely operate within serving queues and GPU allocation policies for LLM backends. SAGE is complementary: it focuses on *pipeline-level* orchestration, treating retrieval, memory, refinement, and generation as co-optimized dataflow stages so that scheduling and placement decisions can be made with an end-to-end view of latency and interference across heterogeneous stages.

**Workflow orchestration and application frameworks.**    Workflow and MLOps platforms such as Airflow and Kubeflow provide scheduling and reproducibility mechanisms (Apache Software Foundation, 2015; Kubeflow, 2018), and lifecycle tools such as MLflow target training and model management (Databricks, 2018). Their execution models commonly assume coarse-grained tasks and batch-oriented performance goals, which complicates interactive inference pipelines with token-level latency metrics and tight SLOs. LLM application frameworks such as LangChain, LlamaIndex, DSPy, and Haystack facilitate rapid construction of RAG and tool-using applications (LangChain, 2023; LlamaIndex , 2023; Khattab et al., 2024; deepset, 2023). In many deployments, performance-critical concerns (tail latency, batching, accelerator sharing, heterogeneous placement) are delegated to external serving and storage services, and the resulting system structure is hard to enforce or evolve at scale. SAGE targets this gap by compiling declarative inference dataflows into distributed execution plans and enforcing a downward-only dependency discipline across layers.

**Retrieval, vector databases, and semantic indexing.**    Vector retrieval is central to RAG pipelines. FAISS provides efficient similarity search primitives (Douze et al., 2025), while vector databases such as Milvus, Weaviate, Chroma, Vespa, and Pinecone offer scalable similarity search and operational features (Wang et al., 2021; Weaviate, 2019; Chroma, 2023; Vespa.ai, 2017; Pinecone Systems, Inc., 2021). These systems are often integrated as standalone services with limited coupling to streaming semantic state maintenance, session memory semantics, or downstream refinement. SAGE integrates vector storage/search and vector-native stream processing as interoperable operators under a unified dataflow model, emphasizing end-to-end pipeline control and measurement rather than isolated retrieval performance.

**Streaming/dataflow systems and incremental computation.**    Foundational dataflow and stream processing systems show the value of stateful operators, windowing, and fault-tolerant incremental computation (Carbone et al., 2015; Zaharia et al., 2013). Timely and Differential Dataflow further emphasize fine-grained incremental maintenance and iterative computation (Murray et al., 2013; McSherry et al., 2013). While general-purpose, these systems motivate SAGE's pipeline-level dataflow abstraction; SAGE specializes the operator set and scheduling targets for LLM-centric inference pipelines.

**Memory systems for LLM applications.**    Many applications require memory beyond a single prompt window, including conversation history, semantic memory over documents, key-value stores, and structured/graph memory. Prior work explores retrieval-based memory augmentation and long-horizon agents (Packer et al., 2023). In practice, developers assemble bespoke memory stacks on top of vector stores or databases. SAGE provides structured memory backends within the system stack so that session semantics and memory access patterns can be expressed and measured as part of the pipeline.

**Context refinement and compression.** Prompt/context compression and refinement reduce token usage while preserving answer quality via learned or heuristic pruning and summarization (Jiang et al., 2023; 2024). These methods are often treated as optional application-level steps. SAGE elevates refinement to a first-class operator so that its compute cost and downstream latency/quality trade-offs can be benchmarked and co-optimized with retrieval and generation.

**Agentic tool use, planning, and evaluation.** Tool use and planning have been studied through prompting and self-supervised approaches such as ReAct and Toolformer (Yao et al., 2023; Schick et al., 2023). Agent evaluations often focus on task success without connecting behaviors to system-level execution control and SLO outcomes. SAGE connects these dimensions by benchmarking agent behaviors (tool selection, planning, timing) alongside system metrics under interference and heterogeneous execution.

**Benchmarking for LLM systems.** MLPerf Inference focuses on inference performance across models and hardware (Reddi et al., 2020; Mattson et al., 2020), while HELM targets broad model quality evaluation (Liang et al., 2023). Agent benchmarks evaluate tool use and multi-step success (Liu et al., 2024; Qin et al., 2024). SAGE contributes a benchmark suite that stresses the coupled pipeline setting, jointly reporting system metrics (throughput, TTFT/TBT, tail latency, SLO compliance, interference) and agent behavior metrics within the same execution substrate.

**Summary.** SAGE differs by elevating end-to-end inference pipelines to first-class declarative dataflows, integrating core inference components within an enforceable five-layer architecture (L1–L5), and providing unified evaluation across both system and agent dimensions.

## B. Architecture Details

SAGE enforces a strict five-layer architecture (L1–L5) with downward-only dependencies to keep the performance-critical runtime independent from domain operators and to allow user-facing tooling to evolve without contaminating the execution path. For clarity, we summarize the responsibilities at the level of architectural layers (rather than enumerating implementation packages).

**Layer responsibilities.** **L1** provides shared foundations (configuration, common types, and portability utilities). **L2** provides platform services (lifecycle, abstract backends, and system-facing adapters). **L3** contains the core dataflow runtime and reusable algorithmic interfaces used by pipelines. **L4** hosts runtime-bound operators that touch heavyweight backends (e.g., memory, vector search, refinement, and inference integration) so their resource usage remains explicit. **L5** provides user-facing interfaces and developer tooling.

## C. Core Runtime Implementation

This section details the lifecycle of a SAGE request, from compilation of a logical dataflow DAG to the dynamic execution loop within each worker replica.

### C.1. Execution Graph Compilation

Compilation transforms a user-defined logical DAG into a physical execution graph ready for distributed deployment. The `ExecutionGraph` in `sage-kernel` performs this lowering by mapping each logical `BaseTransformation` to a set of `TaskNode` replicas determined by the operator's declared parallelism. Each `TaskNode` is paired with a `TaskFactory`, which wraps an `OperatorFactory` responsible for instantiating the operator implementation at runtime.

Edge materialization defines the concrete communication topology. For a logical edge connecting an upstream operator with parallelism $M$ to a downstream operator with parallelism $N$, the compiler materializes explicit physical connections so that fan-out and fan-in are known to the runtime. Outgoing edges from a single replica are grouped by logical downstream target and embedded as routing metadata in the `TaskContext`.

Fan-in is realized by sharing input resources. Each downstream replica is provisioned with a single bounded input channel (via `QueueDescriptor`); multiple upstream replicas are configured to write into this shared channel. For multi-input operators (e.g., joins), physical edges are tagged with explicit input indices, allowing the downstream consumer to demultiplex packets by logical source port even when they arrive on the same physical queue.

---

**Algorithm 1** SAGE Task Worker Loop

---

1: **Input:** $Q_{in}$ (bounded input queue), $Op$ (operator instance), $Router$
2: **Initialize:** $Op$.setup()
3: **while** Running **do**
4:     MAYBE_CHECKPOINT()                                    *// Periodic checkpoint policy*
5:     $Packet \leftarrow Q_{in}$.dequeue_blocking()                  *// Blocks until input arrives*
6:     **if** $Packet$ is STOPSIGNAL **then**
7:         HANDLE_STOP_SIGNAL($Packet$)                       *// Drain/propagate if needed*
8:         **continue**
9:     **end if**
10:     $Op$.receive_packet($Packet$)        *// Sets key from* `partition_key`, *invokes* `process_packet`
11:     UPDATE_METRICS()
12: **end while**

---

### C.2. Worker Loop and Flow Control

Each task replica is driven by a lightweight worker loop that handles data ingestion, operator execution, and output routing. Algorithm 1 formalizes this procedure.

Routing is policy-driven. The `BaseRouter` inspects metadata attached to each `Packet` (e.g., partition keys or strategy hints) and delivers data via one of three strategies: **round-robin** for stateless operators, **broadcast** for control signals or replication, and **hash partitioning** for stateful routing where packets with the same key must reach the same downstream replica.

## D. Reliability and State Management

SAGE distinguishes transient execution state (in-flight packets/items and bounded per-replica input queues) from durable operator state (e.g., session memory or incrementally maintained semantic state). Fault tolerance is aligned with operator semantics via two complementary policies.

### D.1. Checkpoint-based Recovery

Checkpoint-based recovery targets stateful operators where recomputation is costly. Operators expose a uniform state interface via `get_state()` and `restore_state()`. A `CheckpointManager` coordinates snapshot persistence, versioning snapshots by task ID and timestamp. Upon failure, the `Dispatcher` provisions a replacement replica, retrieves the latest valid snapshot, invokes `restore_state()`, and then resumes the worker loop.

### D.2. Restart-based Recovery

For stateless or idempotent operators, SAGE supports lightweight restart policies. The `Dispatcher` can apply fixed-delay or exponential-backoff retries to handle transient errors without snapshot overhead. A heartbeat monitor complements exception handling by detecting silent failures (e.g., timeouts) and triggering the same recovery workflow.

## E. Middleware and Native Backends

This section outlines the L4 middleware layer and how performance-critical components are integrated as dataflow operators.

### E.1. Middleware Component Specifications

The L4 middleware layer is backed by specialized C++ components (built via CMake and exposed as Python operators) so that their resource usage and boundaries remain visible to the runtime. Table 2 summarizes the core components.

*Table 2.* Middleware Component Specifications

| Component | Backend (C++) | Functionality |
|---|---|---|
| sage_db | isage-vdb | Vector storage and similarity search |
| sage_mem | isage-neuromem | Episodic/working memory backends |
| sage_refiner | isage-refiner | Context refinement and compression |
| sage_flow | isage-flow | Vector-native streaming processing |
| sage_tsdb | isage-tsdb | Time-series storage and windowed analytics |

### E.2. Inference Integration via Configurable Backends

SAGE treats generation and embedding as first-class stages by invoking inference through a dedicated operator/function interface (e.g., SageLLMGenerator) rather than embedding model execution inside the core runtime. In our implementation, SageLLMGenerator delegates model execution to a configurable backend created via a factory interface (e.g., local accelerators or remote OpenAI-compatible serving endpoints such as vLLM).

For deployments that require centralized engine management (provisioning, routing, and scheduling across multiple serving instances), SAGE can optionally integrate with an external Control Plane/Gateway (e.g., the independent isagellm package). This separation enables two scaling dimensions: inference engines can scale independently from CPU-bound pipeline orchestration and stateful operators, while preserving a unified execution-plan boundary for observability and policy enforcement.

## F. Interface and Reproducibility

### F.1. Declarative Pipeline DSL

SAGE provides a Python DSL for defining pipelines as declarative dataflows. The runtime compiles the DSL into a physical execution graph and schedules the resulting operators across heterogeneous resources. The example below constructs a simple RAG pipeline.

```python
from sage.kernel.api.local_environment import LocalEnvironment
from sage.libs.foundation.io import FileSource, TerminalSink
from sage.middleware.operators.rag import ChromaRetriever, QAPromptor
from sage.middleware.operators.llm import SageLLMGenerator

env = LocalEnvironment("rag_pipeline")

(
    env.from_source(FileSource, {"file_path": "questions.txt"})
    .map(
        ChromaRetriever,
        {
            "top_k": 5,
            "embedding": {"model": "sentence-transformers/all-MiniLM-L6-v2"},
            "chroma": {
                "collection_name": "kb",
                "persistence_path": ".sage/chroma",
            },
        },
    )
    .map(QAPromptor, {"template": "Answer based on context: {context}\nQ: {query}\nA:"
    })
    .map(SageLLMGenerator, {
        "model_path": "Qwen/Qwen2.5-7B-Instruct",
        "backend_type": "auto",
    })
    .sink(TerminalSink)
)
env.submit()
```

*Listing 1.* Example of SAGE's Declarative DSL for a RAG Pipeline

## F.2. Experimental Hyperparameters (Extended)

*Table 3.* Key Experimental Configuration

| Item | Setting |
|---|---|
| Cluster | 16 CPU nodes (8 cores, 32 GB RAM each), 1 GbE |
| GPU backend | 1 server with NVIDIA A6000 48GB, vLLM (>=0.9.2), `Qwen2.5-3B-Instruct` |
| Embedding | `BAAI/bge-large-en-v1.5` on CPU nodes |
| RAG retriever | FAISS IVF-Flat (1M vectors, 1024-d), top-$k = 5$ |
| Generation | max tokens=1024, temperature=0 (deterministic) |
| Workloads | COMPUTE, RAG, and MIXED (50/50 mixture of Compute and RAG) |

Table 3 summarizes the key settings referenced in the main text. For reproducibility, the full experiment configurations (including workload generation parameters and per-figure sweep grids) are stored as versioned YAML configs under `benchmark/config/`, e.g., `exp_5_1.yaml`, `exp_5_2.yaml`, and `exp_5_3.yaml`. These configs specify the cluster topology, model endpoints, workload mixes (COMPUTE/RAG/MIXED), and the scheduling/admission policies used to produce the figures in §5.

## F.3. Limitations (Extended)

We discuss several limitations observed during evaluation.

**Scheduling overhead at high parallelism.** Certain scheduling policies (LoadAware, Priority) encounter performance degradation at very high parallelism levels (>64 concurrent tasks). This is attributed to the underlying Ray runtime's actor scheduling overhead, which grows superlinearly with task count. Future work could explore native scheduling implementations that bypass Ray's actor model for latency-critical paths.

**Distributed coordination cost.** For fine-grained compute tasks (<10 ms), the overhead of distributed coordination can exceed task execution time. This is a fundamental tension in distributed systems (Amdahl, 1967), not specific to SAGE. Practitioners should ensure task granularity is sufficiently coarse (we recommend >100 ms) to amortize coordination costs.

**Centralized service endpoint bottleneck.** While SAGE's service-oriented design avoids shared-data coordination challenges inherent in distributed systems, routing all requests through centralized service endpoints (e.g., LLM inference server, vector database for retrieval) creates a bottleneck under high task volumes. As concurrent pipelines increase, request queuing and resource contention at these shared services become the dominant performance limiters. Future work could explore request-level load balancing across replicated service instances.

## F.4. Architectural Implications (Extended)

Our evaluation supports the central architectural claim: compiling pipelines into explicit ExecutionGraphs with operator contracts and bounded queues makes design trade-offs *measurable under controlled variation*.

For example, the 11% throughput–46% balance trade-off when swapping FIFO for LoadAware-Spread is predictable because both policies execute the same physical graph. Similarly, the 57% tail latency reduction from admission control directly validates that materializing bounded queues in the execution plan enables explicit backpressure.

The what-if analysis results (<5% prediction error) demonstrate that explicit compilation enables counterfactual reasoning: operators can validate optimization hypotheses before deployment rather than relying on trial-and-error. The automated attribution results (87.5% precision, 84% improvement correlation) show that explicit structure enables active diagnosis and optimization by leveraging critical-path analysis, queue-aware latency decomposition, and constraint-aware recommendations from operator contracts.

Finally, the explainable scheduling results (100% root cause accuracy, 7.8× faster debugging, <2% replay error) highlight that explicit decision traces enable retrospective analysis with counterfactual validation: operators can answer "why this placement?" queries and test alternative policies without redeployment. Overall, making the pipeline the compilation target transforms orchestration from an ad-hoc RPC problem into a controlled systems problem where architectural choices have measurable, attributable, and predictable consequences.

# G. What-If Analysis Mechanism (Extended)

This appendix provides the detailed formalization and mechanism for SAGE's what-if analysis toolkit, which was conceptually introduced in §4.3.

## G.1. Formal Definition of Counterfactual Plans

Given a compiled physical execution graph

$$\mathcal{G}_P = \langle \{v^{(i)}\}, \mathcal{E}, \{Q^{(i)}\}, \pi, \rho \rangle$$

where $\{v^{(i)}\}$ are operator replicas, $\mathcal{E}$ are edges, $\{Q^{(i)}\}$ are bounded input queues, $\pi$ is the placement map from replicas to execution nodes, and $\rho$ is the routing policy over physical edges, the what-if analysis engine constructs a *counterfactual plan* $\mathcal{G}'_P$ by modifying execution-policy parameters, placement, routing, or parallelism while preserving operator semantics and resource constraints.

**Policy Modification Rules.**    The counterfactual plan construction follows these rules:

- **Scheduling policy change**: Modifying $\pi_{\text{sched}}$ (e.g., FIFO $\rightarrow$ LoadAware) produces $\mathcal{G}'_P$ with updated $\pi'_{\text{sched}}$ but identical replicas $\{v^{(i)}\}$ and queue structure $\{Q^{(i)}\}$;

- **Parallelism scaling**: Changing replica count for operator $O_k$ from $n$ to $n'$ modifies both $\{v^{(i)}\}$ (adding/removing replicas) and $\mathcal{E}$ (adjusting edge fanout), while preserving queue capacities and execution policies;

- **Routing strategy change**: Swapping $\rho$ (e.g., RoundRobin $\rightarrow$ KeyHash) preserves the physical topology but alters packet distribution across downstream replicas;

- **Resource reallocation**: Modifying $\pi$ (e.g., moving operators between CPU/GPU nodes) preserves operator logic and parallelism but adjusts node affinity, resource contention, and predicted service times.

## G.2. Trace-Driven Simulation Protocol

The simulator replays historical execution traces (queue depths, service times, blocking durations) under the counterfactual policy configuration. The simulation protocol consists of three phases:

**Phase 1: Trace Extraction.**    From a baseline execution of $\mathcal{G}_P$, the system collects per-replica telemetry:

- *Queue wait time* $T_Q^{(i)}(t)$: Time spent by packet in input queue of replica $v^{(i)}$;

- *Service time* $T_S^{(i)}(t)$: Operator processing time for the packet;

- *Blocking time* $T_B^{(i)}(t)$: Time blocked on downstream queues (backpressure);

- *Routing decisions* $r^{(i)}(t)$: Which downstream replica received each packet.

**Phase 2: Policy Substitution.**    The simulator constructs $\mathcal{G}'_P$ by applying the desired policy change. For scheduling policy changes, the simulator replaces the queue dispatch logic while preserving service time distributions. For parallelism scaling, the simulator adjusts replica counts and rebalances packet distribution according to the modified routing policy.

**Phase 3: Predictive Replay.**    The simulator replays trace events under $\mathcal{G}'_P$, predicting new latencies using queueing models (Theorem H.1). For each packet:

1. Apply routing policy $\rho'$ to determine target replica;

2. Estimate $T'_Q$ using the new queue depth and scheduling policy;

3. Preserve $T_S$ (service time is policy-independent for compute-bound operators);

4. Estimate $T'_B$ using downstream queue occupancy under the new policy;

5. Compute predicted end-to-end latency: $L' = T'_Q + T_S + T'_B$.

### G.3. Accuracy Guarantees and Assumptions

The what-if analysis provides accurate predictions under the following assumptions:

- **Service time stability**: Operator service times are independent of scheduling policy (valid for compute-bound operators, may require calibration for I/O-bound operators with cache effects);

- **Arrival pattern preservation**: The input workload characteristics (request rate, burst patterns) remain constant across baseline and counterfactual scenarios;

- **Resource homogeneity**: Scaling predictions assume homogeneous compute resources (e.g., all replicas run on identical GPU models). Heterogeneity requires per-node calibration.

Empirical validation of prediction accuracy is presented in §I.1, demonstrating $< 5\%$ error for throughput and $< 10\%$ error for tail latency predictions across all tested policy changes.

## H. Queueing-Theoretic Analysis of Backpressure

This appendix provides a formal queueing-theoretic analysis of SAGE's bounded-queue backpressure mechanism, deriving the stability conditions, steady-state behavior, and tail latency bounds stated in Theorem H.1.

**Theorem H.1** (Bounded Buffer Stability). *Consider a pipeline stage with bounded input queue capacity $Q_{\max}$ and service rate $\mu$ (tasks/s). Given an arrival process with rate $\lambda < \mu$, the backpressure mechanism ensures:*

(i) *No unbounded growth*: *Queue occupancy satisfies $\mathbb{E}[Q(t)] \leq Q_{\max}$ for all $t$;*

(ii) *Stable convergence*: *The system reaches steady state with bounded mean queue length $\mathbb{E}[Q_\infty] = \frac{\rho}{1-\rho}$ where $\rho = \lambda/\mu < 1$;*

(iii) *Tail latency bound*: *Under Poisson arrivals, P99 latency is bounded by $W_{0.99} \leq \frac{-\ln(0.01)}{\mu(1-\rho)} + \frac{1}{\mu}$.*

**Modeling Assumptions vs. Implementation Properties.** The bounded-queue stability analysis relies on two types of guarantees: (1) *architectural invariants* enforced by the runtime implementation (bounded capacity $Q_{\max}$, FIFO discipline, blocking backpressure), and (2) *workload assumptions* used in the queueing-theoretic model (Poisson arrivals, exponential service times). While the runtime strictly enforces (1) via `Queue(maxsize=...)` with blocking `get()` semantics (verified in `base_task.py:383`), real-world LLM workloads exhibit bursty arrivals and heavy-tailed inference latency, causing the M/M/1 model to underestimate tail behavior by 6–8% (§ I.1). The following analysis provides a *first-order approximation* that captures steady-state behavior in the stability region ($\rho < 1$), with experimental validation demonstrating sufficient accuracy for pre-deployment policy optimization.

### H.1. System Model

We model a pipeline stage as a single-server queue with the following characteristics:

**Arrival Process.** Tasks arrive according to a Poisson process with rate $\lambda$ (tasks/s). This assumption is standard in queueing theory and approximates many practical workloads under independence (Kleinrock, 1975).

**Service Process.** The operator processes tasks with exponentially distributed service times, mean $1/\mu$ (seconds/task). The service rate $\mu$ represents the operator's inherent processing capacity.

**Queue Discipline.** SAGE enforces a **bounded FIFO queue** with capacity $Q_{\max}$. When the queue is full, upstream writes block ; this constitutes the backpressure mechanism.

**Traffic Intensity.** We define the traffic intensity (utilization) as:

$$\rho = \frac{\lambda}{\mu} \tag{5}$$

The system is stable when $\rho < 1$ (arrival rate less than service rate).

## H.2. Proof of Theorem H.1

We now prove each component of Theorem H.1.

### H.2.1. (I) NO UNBOUNDED GROWTH

**Claim:** Queue occupancy satisfies $Q(t) \leq Q_{\max}$ for all $t \geq 0$.

**Proof:** By construction, the bounded queue enforces $Q(t) \leq Q_{\max}$ through the blocking write primitive in Algorithm 1:

```
if Q.size() >= Q_max:
    block_until(Q.size() < Q_max)
Q.enqueue(packet)
```

This ensures that *no task can be enqueued when* $Q(t) = Q_{\max}$. Therefore, $Q(t) \in [0, Q_{\max}]$ for all $t$, eliminating buffer overflow by construction. Taking expectations:

$$\mathbb{E}[Q(t)] \leq Q_{\max} \quad \forall t \geq 0 \tag{6}$$

□

### H.2.2. (II) STABLE CONVERGENCE

**Claim:** When $\rho < 1$, the system converges to a steady state with mean queue length:

$$\mathbb{E}[Q_\infty] = \frac{\rho}{1 - \rho} \tag{7}$$

**Proof:** For $\rho < 1$, the bounded queue behaves as an **M/M/1 queue** in the stable region. The steady-state analysis proceeds as follows:

**Balance Equations.** Let $\pi_n$ denote the steady-state probability that $n$ tasks are in the system. The global balance equations for an M/M/1 queue are:

$$\lambda \pi_0 = \mu \pi_1 \tag{8}$$
$$(\lambda + \mu)\pi_n = \lambda \pi_{n-1} + \mu \pi_{n+1} \quad \text{for } n \geq 1 \tag{9}$$

Solving recursively yields:

$$\pi_n = (1 - \rho)\rho^n \quad \text{for } n \geq 0 \tag{10}$$

**Mean Queue Length.** The expected number of tasks in the system is:

$$\mathbb{E}[N] = \sum_{n=0}^{\infty} n\pi_n = \sum_{n=0}^{\infty} n(1 - \rho)\rho^n = \frac{\rho}{1 - \rho} \tag{11}$$

**Little's Law.** By Little's Law (Little, 1961), the mean queue length $\mathbb{E}[Q_\infty]$ relates to the mean waiting time $\mathbb{E}[W]$ as:

$$\mathbb{E}[Q_\infty] = \lambda \cdot \mathbb{E}[W] = \frac{\rho}{1 - \rho} \tag{12}$$

This confirms that the system converges to a stable steady state when $\rho < 1$. □

### H.2.3. (III) TAIL LATENCY BOUND

**Claim:** Under Poisson arrivals, the 99th percentile latency is bounded by:

$$W_{0.99} \leq \frac{-\ln(0.01)}{\mu(1 - \rho)} + \frac{1}{\mu} \tag{13}$$

**Proof:** In an M/M/1 queue, the waiting time $W$ (queueing delay + service time) follows an exponential distribution:

$$P(W > w) = e^{-\mu(1-\rho)w} \tag{14}$$

To find the 99th percentile $W_{0.99}$ where $P(W > W_{0.99}) = 0.01$, we solve:

$$e^{-\mu(1-\rho)W_{0.99}} = 0.01 \tag{15}$$

$$-\mu(1-\rho)W_{0.99} = \ln(0.01) \tag{16}$$

$$W_{0.99} = \frac{-\ln(0.01)}{\mu(1-\rho)} \tag{17}$$

Adding the mean service time $1/\mu$ (since $W$ includes service time), we obtain:

$$W_{0.99} = \frac{-\ln(0.01)}{\mu(1-\rho)} + \frac{1}{\mu} \approx \frac{4.605}{\mu(1-\rho)} + \frac{1}{\mu} \tag{18}$$

This provides an **analytical upper bound** on tail latency as a function of system utilization $\rho$. $\square$

### H.3. Backpressure Propagation Analysis

When a downstream queue saturates ($Q_{\text{down}}(t) = Q_{\max}$), backpressure propagates upstream through blocking writes. We analyze the transient dynamics.

**Propagation Time.** Consider a linear pipeline with $K$ stages, each with service rate $\mu$ and queue capacity $Q_{\max}$. When stage $k$ saturates, stage $k-1$ experiences blocked writes after a propagation delay $\Delta t_k$.

**Upper bound on propagation time:** The worst-case propagation delay from stage $K$ to stage 1 is bounded by:

$$T_{\text{prop}} \leq \sum_{k=1}^{K} \frac{Q_{\max}}{\mu_k} \tag{19}$$

where $\mu_k$ is the service rate of stage $k$. This represents the time to drain all queues sequentially.

**Cascade Prevention.** Bounded queues guarantee that backpressure *stabilizes the system globally*:

- **Without bounded queues:** A saturated downstream stage causes unbounded queue growth upstream, leading to memory exhaustion (cascading failure).

- **With bounded queues:** Blocked writes prevent queue growth, and the system enters a *stable blocking state* where throughput matches the bottleneck's service rate: $\lambda_{\text{eff}} = \min_k \mu_k$.

### H.4. Multi-Stage Pipeline Analysis

For a pipeline with $K$ stages in series, each modeled as M/M/1:

**End-to-End Latency.** The total latency is the sum of per-stage latencies:

$$W_{\text{total}} = \sum_{k=1}^{K} W_k = \sum_{k=1}^{K} \frac{1}{\mu_k(1-\rho_k)} \tag{20}$$

where $\rho_k = \lambda/\mu_k$ is the utilization of stage $k$.

**Bottleneck Stage.** The stage with maximum utilization determines system throughput:

$$\lambda_{\max} = \min_k \mu_k \tag{21}$$

For stability, we require $\lambda < \lambda_{\max}$.

**Admission Control Implication.** Theorem H.1 implies an admission control strategy: reject arrivals when $\lambda$ approaches $\lambda_{\max}$ to prevent the system from entering the unstable region ($\rho \to 1$).

### H.5. Experimental Validation

The experimental results in § 5.5 validate these theoretical predictions:

**Stability under load.** Figure 7 shows that per-job efficiency degrades gracefully from 100% to 71% as concurrent jobs increase from 1 to 8. This matches the prediction from Theorem H.1(ii): bounded queues prevent unbounded growth even under contention.

**Tail latency control.** Figure 6 demonstrates that staggered admission reduces P99 latency by 57% (77 s $\to$ 33 s). This aligns with equation (12): reducing effective arrival rate $\lambda$ lowers utilization $\rho$, directly decreasing $W_{0.99}$.

**Quantitative comparison.** Consider a stage with $\mu = 10$ tasks/s and $Q_{\max} = 100$:

- At $\lambda = 8$ tasks/s ($\rho = 0.8$): Predicted $W_{0.99} = \frac{4.605}{10 \times 0.2} + 0.1 = 2.40$ s

- At $\lambda = 5$ tasks/s ($\rho = 0.5$): Predicted $W_{0.99} = \frac{4.605}{10 \times 0.5} + 0.1 = 1.02$ s

The 57% reduction observed experimentally (77 s $\to$ 33 s) corresponds to a similar proportional decrease, validating the model.

### H.6. Design Implications

The queueing-theoretic analysis has direct architectural consequences:

**Queue sizing.** $Q_{\max}$ should be chosen to accommodate transient bursts without triggering excessive backpressure:

$$Q_{\max} \geq \frac{\lambda_{\text{peak}} - \mu}{\mu} \cdot T_{\text{burst}} \tag{22}$$

where $T_{\text{burst}}$ is the expected burst duration.

**Overprovisioning.** To maintain P99 latency below target $W_{\text{target}}$, the service rate must satisfy:

$$\mu \geq \lambda \left( 1 + \frac{4.605}{\mu \cdot W_{\text{target}}} \right) \tag{23}$$

**Admission control threshold.** Reject arrivals when observed utilization exceeds a safety margin:

$$\hat{\rho}_{\text{observed}} > \rho_{\text{threshold}} = 0.85 \tag{24}$$

This prevents the system from entering the high-latency regime where $W_{0.99}$ explodes.

### H.7. Limitations and Extensions

**Beyond M/M/1.** Our analysis assumes Poisson arrivals and exponential service times. For non-Markovian workloads (e.g., heavy-tailed service times), the bounds become more conservative but the stability condition $\lambda < \mu$ remains valid by Kingman's bound (Kingman, 1962).

**Priority scheduling.** For priority queues, the analysis extends to M/M/1 with priority classes, where high-priority traffic maintains lower latency at the expense of low-priority tail latency.

**Network effects.** In distributed settings, cross-node communication latency adds to service time, increasing effective $1/\mu$. The model extends naturally by treating network latency as part of the service distribution.

### H.8. Summary

This appendix establishes that SAGE's bounded-queue design provides *provable stability guarantees* under standard queueing assumptions. Theorem H.1 quantifies the conditions ($\rho < 1$) and bounds (tail latency) that enable predictable behavior—a property experimentally validated in § 5. By making backpressure *explicit and bounded*, SAGE transforms LLM pipeline orchestration from an ad-hoc coordination problem into a *formally analyzable queueing system*.

## I. Additional Evaluation: What-If, Attribution, and Explainability

This section provides extended evaluation details for three advanced features: what-if analysis, performance attribution, and explainable scheduling. These features demonstrate SAGE's diagnostic and predictive capabilities beyond core execution.

### I.1. Validating What-If Analysis Predictions

We evaluate the what-if analysis toolkit (§ 4.3) by comparing predicted vs. measured metrics under controlled changes in (i) parallelism, (ii) scheduling policy, and (iii) routing strategy on the RAG pipeline.

**Methodology.** For each what-if query, we: (1) collect 5-minute execution traces under baseline configuration, (2) generate counterfactual ExecutionGraph $\mathcal{G}'_P$ with modified policy parameters, (3) use trace-driven simulation to predict throughput and P99 latency, and (4) deploy the actual configuration to measure ground truth. Prediction error is computed as

$$\text{Error} = \frac{|\text{Predicted} - \text{Actual}|}{\text{Actual}}$$

*Table 4.* What-If Prediction Accuracy across Three Modification Scenarios on the RAG Pipeline

| Scenario | Metric | Predicted | Actual | Error |
|---|---|---|---|---|
| **Parallelism** | Throughput (tasks/s) | 14.2 | 13.8 | 2.9% |
| $4 \rightarrow 8$ workers | P99 Latency (s) | 3.8 | 4.1 | 7.9% |
| **Scheduler** | Throughput (tasks/s) | 16.8 | 17.1 | 1.8% |
| FIFO $\rightarrow$ LoadAware | P99 Latency (s) | 3.2 | 3.4 | 6.3% |
| **Routing** | Throughput (tasks/s) | 17.3 | 16.9 | 2.4% |
| RoundRobin $\rightarrow$ KeyHash | P99 Latency (s) | 3.1 | 3.3 | 6.5% |

Table 4 shows prediction errors ranging from **1.8% to 7.9%**, with mean absolute error of **4.6%**. The predictions are sufficiently accurate to distinguish meaningful performance differences: for instance, the predicted 19% throughput gain (14.2→17.1 tasks/s) when switching from FIFO to LoadAware aligns with the actual 24% improvement, correctly identifying LoadAware as superior for load-balanced workloads.

**Error sources.** The 6–8% errors in P99 latency predictions primarily arise from two factors: (1) trace replay cannot perfectly capture bursty arrival patterns or stochastic LLM inference latency, and (2) the queueing model (Theorem H.1) assumes Poisson arrivals, which underestimates tail behavior under correlated workloads. Despite these limitations, predictions remain within 10% of ground truth, accurate enough for pre-deployment validation.

**Architectural Validation:** These results confirm the core utility claim: what-if analysis transforms policy optimization from *trial-and-error deployment* to *evidence-based decision-making*. The ability to predict trade-offs with $< 5\%$ error stems directly from SAGE's explicit ExecutionGraph representation ; whereas RPC-based systems cannot support counterfactual reasoning because service boundaries and policy are implementation-entangled. This capability reinforces the central thesis: making pipelines a first-class compilation target enables testable, attributable orchestration decisions.

### I.2. Validating Automated Performance Attribution

We evaluate the automated attribution system (§ 4.4) by injecting known performance issues into three pipeline configurations and measuring whether the method identifies the correct bottleneck operator, classifies the root cause, and produces actionable optimization recommendations.

**Methodology.** We construct three synthetic scenarios with deliberate bottlenecks: (1) **Compute-bound**: RAG pipeline with artificially inflated embedding computation time (100 ms $\rightarrow$ 500 ms), (2) **I/O-bound**: RAG pipeline with downstream LLM generator limited to 1 worker (creating backpressure), and (3) **Scheduling overhead**: Compute pipeline with excessive parallelism (64 workers) causing dispatch saturation. For each scenario, we: (1) collect 5-minute execution traces, (2) run automated attribution to identify bottlenecks and classify root causes, and (3) apply recommended optimizations to measure actual performance improvement.

*Table 5.* Automated Attribution Accuracy across Three Synthetic Bottleneck Scenarios

| Scenario | Identified Bottleneck | Classification | Recommendation | Improvement |
|---|---|---|---|---|
| Compute-bound (slow embedding) | Embedding (82% latency) | Compute-bound | Scale 4$\rightarrow$8 workers | +73% throughput |
| I/O-bound (LLM backpressure) | Retriever (68% blocking) | I/O-bound | Scale LLM 1$\rightarrow$4 workers | -54% P99 latency |
| Scheduling overhead (64 workers) | Dispatcher (91% queue wait) | Scheduling overhead | Reduce to 16 workers | +42% throughput |

Table 5 shows that the system correctly identified the bottleneck operator in all three cases, with accurate root cause classification. The recommended optimizations yielded substantial improvements: **+73% throughput** for compute-bound (scaling embedding workers), **-54% P99 latency** for I/O-bound (scaling downstream LLM generator), and **+42% throughput** for scheduling overhead (reducing over-parallelization).

**False Positive Analysis.** In a separate experiment with 12 production RAG pipelines (no deliberate bottlenecks), the attribution system flagged 8 operators as potential bottlenecks. Manual inspection confirmed 7 were legitimate (87.5% precision), with 1 false positive due to bursty arrival patterns misclassified as compute-bound. The system correctly avoided flagging 42 non-bottleneck operators (no false negatives observed).

**Recommendation Quality.** We evaluate recommendation quality by measuring the correlation between predicted improvement (via what-if analysis) and actual improvement after applying recommendations. Across 15 bottleneck scenarios, the predicted vs. actual improvement showed strong correlation ($R^2 = 0.84$), with mean absolute error of 12%. This validates that the integration of attribution and what-if analysis produces *reliable, actionable guidance* rather than speculative suggestions.

**Architectural Validation:** These results demonstrate that automated attribution transforms observability from *passive monitoring* to *active diagnosis*. The 87.5% precision and 84% prediction correlation stem from SAGE's explicit representation of execution structure: critical path analysis requires a concrete DAG, latency decomposition requires per-replica queue metrics, and constraint-aware recommendations require operator contracts. RPC-based systems cannot automate attribution because they lack the structural and telemetry foundations , where service boundaries hide cross-stage dependencies, and missing observability hooks prevent isolating queue/service/blocking components. This validates the core thesis: making pipelines first-class compilation targets enables not just measurement, but *automated understanding* of performance behavior.

### I.3. Explainable Scheduling Mechanism (Extended)

This subsection contains the detailed mechanism description that is compressed in the main text (§ 4.7).

**Decision Explanation and Trace Reconstruction.** For each placement decision $\pi(v^{(i)}) = n_j$, the scheduler records a structured *decision trace* $\mathcal{D}$ capturing viable candidates under constraints, the filters that prune candidates, the policy-specific scoring function, and the final selection with quantitative scores for alternatives.

**Policy Comparison via Strategy Profiling.** Given execution traces under policy $\pi_{\text{sched}}^{A}$, SAGE replays decisions under an alternative policy $\pi_{\text{sched}}^{B}$ using recorded operator contracts and node state snapshots to compare resource balance, latency impact (via queueing analysis, Theorem H.1), and placement churn.

**Replay Debugging for Tail Latency.** When tail latency violations occur, SAGE can isolate anomalous intervals, reconstruct execution context (placements, queue states, and active policies), identify causal factors, and simulate alternative policies/placements to validate mitigations quantitatively.

**Architectural Enablers.** Explainability relies on SAGE's explicit `ExecutionGraph` and operator contracts: placement decisions are first-class and traceable, constraints and preferences are explicit, and bounded queues expose queue depth as a load signal. These ingredients are typically implicit or opaque under RPC composition.

## I.4. Validating Explainable Scheduling

We evaluate the explainable scheduling framework (§ 4.7) through three experiments: (1) decision trace completeness and interpretability, (2) policy comparison accuracy via policy-swap replay, and (3) tail latency replay fidelity.

**Decision Explanation Completeness.** We evaluate whether decision traces $\mathcal{D}$ capture sufficient information to answer counterfactual queries. For each placement decision $\pi(v^{(i)}) = n_j$ across 500 scheduling events (spanning FIFO, LoadAware, and Priority policies under mixed RAG workloads), we verify: (1) **Constraint coverage**: Are all resource constraints (CPU/GPU affinity, memory limits) recorded? (2) **Scoring transparency**: Can users reconstruct utility scores $U(n)$ for all viable nodes? (3) **Alternative visibility**: Are rejected candidates and their rejection reasons logged?

*Table 6.* Decision Explanation Quality Metrics across 500 Scheduling Events

| Quality Metric | FIFO | LoadAware | Priority |
|---|---|---|---|
| Constraint coverage (complete traces) | 100% | 100% | 98.6% |
| Scoring transparency (reproducible $U(n)$) | 100% | 97.3% | 95.8% |
| Alternative visibility (rejected nodes logged) | N/A | 96.1% | 94.2% |
| Explanation generation time (ms/event) | 0.8 | 1.4 | 1.9 |

Table 6 shows high completeness: **98.6–100% constraint coverage**, **95.8–100% scoring transparency**, and **94.2–96.1% alternative visibility** (FIFO has no alternatives since it uses first-available placement). The 2.7% gap in LoadAware scoring transparency stems from queue depth sampling latency: when queue depths change between decision time and trace recording ($< 5$ ms window), scores become slightly stale but remain within 10% of true values. Explanation generation overhead is negligible (**0.8–1.9 ms per event**), representing $< 0.5\%$ of typical scheduling latency (300–500 ms).

**User Study: Interpretability.** We conduct a small user study with 8 system operators (4 experienced, 4 novice) to assess whether explanations are *actionable*. Participants receive decision traces for 10 anomalous placements (e.g., "Why was GPU operator placed on CPU node?") and are asked to identify root causes within 5 minutes per case. Results show **92.5% correct identification rate** (74/80 cases), with experienced operators achieving 95% accuracy vs. 90% for novices. Participants report that structured traces with quantitative scores (e.g., "$U(n_3) = -85$ vs. $U(n_4) = -120$") enable rapid hypothesis testing, compared to unstructured log analysis (baseline condition with raw scheduler logs yielded 68% accuracy).

**Policy Comparison via Strategy Profiling.** We measure whether profiling accurately predicts performance shifts when swapping policies. We run production traces (2 hours, mixed RAG workload on 16 nodes) under FIFO, replay them under LoadAware policy, and compare predicted vs. actual metrics:

*Table 7.* Predicted vs. Actual Performance under Policy Swap (FIFO $\rightarrow$ LoadAware)

| Metric | FIFO (actual) | LoadAware (predicted) | LoadAware (actual) | Error |
|---|---|---|---|---|
| Throughput (tasks/s) | 12.34 | 10.98 | 11.21 | 2.1% |
| P99 latency (ms) | 842 | 658 | 673 | 2.2% |
| Gini coefficient | 0.51 | 0.28 | 0.31 | 10.7% |
| Placement churn (%/cycle) | 0.8% | 4.5% | 4.8% | 6.7% |

Table 7 shows strong agreement: **throughput and latency predictions are within 2.2%**, while load balance (Gini coefficient) shows larger error (10.7%) due to queueing model simplifications (we assume M/M/1 behavior, but actual service times exhibit heavy-tailed distributions). Predicted placement churn (4.5%) closely matches actual (4.8%), validating that replay captures dynamic rebalancing overhead.

**Replay Debugging for Tail Latency.** To validate replay fidelity, we inject a tail latency anomaly: at $t = 30$ min, we artificially slow down 2 nodes (reducing CPU capacity by 50%) for 5 minutes, causing P99 latency to spike from 650 ms to 1250 ms. We then use replay debugging to reconstruct the incident.

The debugger correctly: (1) **Isolated the anomalous interval** ($t \in [1800, 2100]$ s) using CUSUM anomaly detection, (2) **Identified causal factors**: "LoadAware policy failed to redistribute load because queue depth signals lagged by $\sim 10$ s, causing 42% of tasks to route to degraded nodes", and (3) **Simulated alternatives**: Switching to a faster-reacting LoadAware variant (1 s queue sampling instead of 10 s) would have reduced P99 to 780 ms (actual re-run confirms 795 ms, **1.9% error**).

**Comparison with Baseline Debugging.** We compare explainable scheduling against two baselines: (1) **Manual log analysis**: Operators inspect raw scheduler logs to diagnose the same tail latency incident, and (2) **APM tools**: Commercial Application Performance Monitoring tools (e.g., Datadog, New Relic) that provide service-level metrics but lack scheduling-aware attribution.

*Table 8.* Debugging Efficiency Comparison for Tail-Latency Incident Analysis

| Approach | Time to Diagnosis | Root Cause Accuracy | Mitigation Validation |
|---|---|---|---|
| Manual log analysis | 47 min | 62.5% | Not feasible |
| APM tools (Datadog) | 23 min | 75.0% | Qualitative only |
| Explainable scheduling | **6 min** | **100%** | **Quantitative (1.9% error)** |

Table 8 shows that explainable scheduling achieves **7.8× faster diagnosis** than manual logs (6 min vs. 47 min), **3.8× faster than APM tools** (6 min vs. 23 min), and provides **100% root cause accuracy** vs. 62.5–75% for baselines. Crucially, only explainable scheduling enables *quantitative mitigation validation* (simulating alternative policies with $<2$ % error), while baselines rely on trial-and-error redeployment.

**Architectural Validation:** These results demonstrate that explainable scheduling transforms debugging from *reactive log inspection* to *proactive root-cause analysis with counterfactual validation*. The 100% root cause accuracy and $< 2$% replay error stem from SAGE's explicit `ExecutionGraph` and decision trace design: structured placement decisions enable reconstruction of "why" (not just "what"), queueing models enable predictive replay, and operator contracts enable constraint-aware alternative generation. RPC-based systems cannot support explainable scheduling because placement decisions are implicit in service discovery (e.g., DNS round-robin, load balancer heuristics) and lack the structured traces required for counterfactual reasoning. This validates that *making scheduling decisions explicit and traceable* is an architectural prerequisite for automated incident analysis in multi-stage pipelines.

## J. Automated Performance Attribution (Extended)

This section provides the full description of SAGE's automated performance attribution pipeline that is compressed in the main text (§ 4.4). The attribution engine operates on execution traces over $\mathcal{G}_P$ and decomposes end-to-end latency into per-operator *queue wait*, *service*, and *blocking* components. It identifies bottlenecks using a critical-path style traversal, classifies dominant constraints (compute-bound, downstream backpressure/IO-bound, or scheduling/dispatch overhead), and generates targeted recommendations (e.g., scale a replica, rebalance downstream capacity, tune bounded queues, or switch scheduling policy). The what-if toolkit (§ 4.3) is used to rank actions by predicted impact.

