# OpenReview forum: "SAGE: A Dataflow-Native Framework for Modular, Controllable, and Transparent LLM-Augmented Reasoning"
_ICML.cc/2026/Conference — ICML 2026 regular_

### Official Review · Reviewer_ctdS · 2026-03-03

**Soundness:** 3
**Presentation:** 2
**Significance:** 2
**Originality:** 2
**Overall Recommendation:** 3
**Confidence:** 4

**Summary:**

This paper presents SAGE, a framework that treats multi-stage LLM inference pipelines as a compilation task. Each operator composes a node in DAG, and A compiler lowers the logical DAG into a compute graph with backpressure control, replicas, and communication channels. The system includes what-if analysis for evidence-based decision-making. On a 16-node CPU, SAGE reports 11.4× throughput on RAG workloads and 57% P99 latency reduction.

**Compliance With Llm Reviewing Policy:**

Affirmed.

**Final Justification:**

The paper's title mentions reasoning, but the evaluation is limited to relatively fixed workloads (self-defined COMPUTE and rag). A natural concern is that dynamic reasoning paths may introduce different datapath and break the pipeline. Moreover, pipeline construction (asynchronous execution) is supported by LangChain, LlamaIndex, and Ray Serve. The claim of "First pipeline-as-compilation-target abstraction" is a bit overstated. Based on these reasons, I'd like to keep my score.

**Key Questions For Authors:**

1. How does SAGE handle dynamic control flow (runtime branching, loops, conditional tool calls) required by agentic workloads? If it cannot, the scope claim should be narrowed.
2. The COMPUTE workload is used in multiple experiments but never defined. What does it compute, and what is the per-task latency?

**Limitations:**

yes

**Strengths And Weaknesses:**

**Strengths:**
1. Treating the pipeline as a compilation target is well-motivated. By requiring each operator to declare its resource needs, state, and I/O behavior, the system can reason about the entire pipeline.
2. The five-layer architecture cleanly separates logical specification from physical execution. The paper is well-structured overall.
3. The gap between LLM serving engines is real. A principled pipeline-level abstraction addresses a practical need.

**Weaknesses:**
1. SAGE is only compared against its own variants (FIFO, RoundRobin, LoadAware, Priority), with no baseline against existing systems.
2. The COMPUTE workload is never concretely defined: no specification of what it computes, input/output sizes, or latency distribution.
3. SAGE compiles static DAGs, yet the title highlights agentic reasoning (ReAct, tool-use, iterative planning). Agents decide at runtime which tools to call and how many rounds to iterate. The paper does not evaluate any agentic workload.
4. The paper over-formalizes straightforward engineering decisions as theorems, which obscures rather than clarifies the contribution.

---

> ### Author Rebuttal · Authors · 2026-03-31
>
> ## Response: On dynamic control flow and "agentic reasoning"
>
> *[Addresses: Weakness 3; Key Question 1]*
>
> We agree that the current submission does not evaluate a fully general agentic workload, and the title/introduction overstated that scope. SAGE currently supports a structured subset of dynamic control flow: `FilterOperator` for data-dependent routing, and `FutureOperator` + `fill_future` for bounded feedback loops. At system level, pipeline-as-a-service plus queue-based inter-pipeline messaging supports orchestration-operator patterns across tool pipelines.
>
> In revision, we will narrow claims to what is demonstrated (structured multi-stage inference pipelines), explicitly state that fully general agent runtimes are not evaluated, and add an appendix documenting supported control-flow patterns, deterministic vs. non-deterministic guarantee boundaries, and operator-level diagnostics for latency-variance-heavy stages.
>
> ---
>
> ## Response: On the COMPUTE workload definition
>
> *[Addresses: Weakness 2; Key Question 2]*
>
> We agree COMPUTE was under-specified and will add a reproducible workload definition (topology, payload, operator logic, and measurement protocol).
>
> COMPUTE is a synthetic CPU-bound pipeline (`Source -> ComputeOp -> Sink`) used to isolate orchestration overhead from model inference. Each request is a JSON payload (~200 tokens, ~800 bytes). `ComputeOp` runs a deterministic 100K-iteration busy loop and returns the payload unchanged. No embedding/retrieval/LLM call is involved.
>
> We will report operator-level and end-to-end latency with units/percentiles. Current measurements: operator mean 3.52 ms (P50 3.49, P99 3.91); end-to-end at single-node concurrency 1 mean 26.4 ms (P50 25.1, P99 39.7). This clarifies Fig. 3: COMPUTE is coordination-dominated, while RAG amortizes per-hop orchestration cost with longer service time. MIXED is a 50/50 COMPUTE/RAG interleaving.
>
> ---
>
> ## Response: On the lack of external baselines
>
> *[Addresses: Weakness 1]*
>
> We agree internal-only comparisons are insufficient. We added external baselines on a matched RAG setup (same vLLM backend, FAISS 1M vectors, query set, and `max_tokens=20`): LangChain v0.3, LlamaIndex v0.11, and Ray Serve v2.40.
>
> Key results: under concurrency scaling, SAGE reaches 18.0 req/s at concurrency 8 versus 9.07 (LlamaIndex), 6.76 (LangChain), and 2.51 (Ray Serve). In multi-pipeline contention, SAGE keeps 87% throughput retention at 4 pipelines (11.0/12.7), versus 85% (LlamaIndex) and 29% (Ray Serve), while maintaining higher absolute throughput than both LangChain and LlamaIndex.
>
> We will include full baseline tables in the revision and clearly separate their role from internal policy ablations: external baselines establish competitive positioning; FIFO/RoundRobin/LoadAware/Priority isolate policy effects on a fixed graph. The baseline protocol is identical across our rebuttal responses to ensure a single consistent comparison setup.
>
> ---
>
> ## Response: On over-formalization
>
> *[Addresses: Weakness 4]*
>
> We agree §3 is over-formalized in presentation. In revision, we will lead with a concrete compilation walkthrough and move notation after intuition. The formal elements will be framed as specification tools (for reproducibility and fair policy comparison), not as standalone theory claims.

---

> > ### Author Rebuttal · Reviewer_ctdS · 2026-04-03
> >
> > Your rebuttal addresses my concerns. However, most LLM reasoning is dynamic, while SAGE uses a static DAG. I’d like to keep my score.

---

> > > ### Author Response · Authors · 2026-04-04
> > >
> > > Thank you for the acknowledgement.
> > >
> > > We believe the remaining concern rests on an overly strong equation between **agentic behavior** and **fully runtime-generated execution**, whereas SAGE is designed for **structured agentic execution** in which the workflow remains compilable but runtime behavior is still dynamic through controller decisions, branching, tool use, and bounded refinement.
> > >
> > > This is a practically important regime, reflected in systems such as:
> > >
> > > - **OpenClaw**, which uses explicit skills and workflows for assistants;
> > > - **Claude Code**, which runs an agentic loop over tools and subagents; and
> > > - **AWS-style SRE assistants**, where incident response remains grounded in explicit tool interfaces and operational workflows even though investigation and action selection are dynamic.
> > >
> > > Since this is exactly the class of agentic execution SAGE is designed to support, we respectfully hope that you reconsider this point and raise your score accordingly.

---

### Official Review · Reviewer_tHWN · 2026-03-07

**Soundness:** 3
**Presentation:** 3
**Significance:** 3
**Originality:** 3
**Overall Recommendation:** 4
**Confidence:** 4

**Summary:**

This paper presents SAGE (Streaming-Augmented Generative Execution), a system that treats multi-stage LLM inference pipelines as first-class compilation targets rather than ad-hoc compositions of RPC-connected services. The core idea is a declarative dataflow model where each pipeline stage (retrieval, embedding, refinement, generation, memory, tool use) is an operator with explicit contracts specifying resource requirements, state semantics, and I/O behavior. A compiler lowers logical pipeline DAGs into physical execution graphs with bounded-queue backpressure, replica placement, and routing policies. The system is evaluated on a 16-node CPU cluster with a single GPU server, demonstrating 11.4x throughput scaling for RAG workloads, a 57% P99 latency reduction via admission control, and pluggable scheduling policies that yield measurable throughput-balance trade-offs. Additional contributions include what-if counterfactual analysis, automated performance attribution, and explainable scheduling.

**Compliance With Llm Reviewing Policy:**

Affirmed.

**Final Justification:**

SAGE presents a well-motivated system that treats multi-stage LLM inference pipelines as first-class compilation targets rather than ad-hoc RPC compositions. The core abstraction, a declarative dataflow model with explicit operator contracts compiled into physical execution graphs, is a principled contribution that enables controlled policy comparison, what-if analysis, and operator-level attribution on a fixed graph structure. The rebuttal substantially addressed the most critical gap by adding external baselines against LangChain, LlamaIndex, and Ray Serve, demonstrating meaningful throughput advantages (2x over LlamaIndex, 2.7x over LangChain) on matched RAG workloads. The authors also agreed to soften overclaimed novelty language and reposition the work more precisely relative to prior dataflow systems. However, several concerns remain partially addressed: the evaluation uses only a 3B parameter model on a single GPU, leaving open whether pipeline-level optimizations matter when generation dominates at production scale; the contract misspecification degradation curve was acknowledged as missing but not provided; and the what-if analysis failure modes near decision boundaries received only anecdotal rather than systematic treatment. The paper's formal contributions, particularly Theorem 3.1, are better understood as specification results than deep theoretical advances, which the authors acknowledged. The writing and appendix material are thorough, and the experimental methodology of swapping policies on a fixed execution graph is a genuinely useful design for isolating scheduling effects. Overall, this is a technically solid systems paper with a clean abstraction and good evaluation methodology, though the limited hardware scale and deferred sensitivity analyses prevent full confidence in the generalizability of the results; I maintain my score of weak accept.

**Key Questions For Authors:**

1. Can you provide a direct comparison against at least one existing pipeline orchestration baseline (e.g., Ray Serve, a LangChain/LlamaIndex pipeline, or a hand-tuned RPC composition) on the same workloads? Without this, it is difficult to assess whether SAGE's abstractions provide tangible benefits beyond what competent engineering with existing tools achieves. A positive comparison here would significantly strengthen the paper.

2. How does SAGE perform with realistically sized models (e.g., 70B+ parameters) on multi-GPU deployments where KV-cache management, prefill/decode disaggregation, and GPU memory pressure become dominant? The current evaluation with a 3B model on a single GPU leaves open whether the pipeline-level optimizations matter when the generation stage itself is the overwhelming bottleneck. Evidence at larger scale would raise my confidence in significance.

3. The operator contracts are currently user-specified, and Section 5.9 acknowledges that inaccurate contracts lead to suboptimal decisions. Have you measured the sensitivity of SAGE's compilation and scheduling quality to contract accuracy? For instance, what happens when memory or GPU requirements are misspecified by 2x or 5x? Understanding this degradation curve is important for practical adoption.

4. Theorem 3.1 assumes deterministic operators for order preservation. How does SAGE handle non-deterministic operators (e.g., LLM generation with temperature > 0, or tool calls with variable latency) in practice? Does the system provide any guarantees or diagnostics for pipelines with such operators?

5. The what-if analysis reports < 5% throughput prediction error, but the P99 latency errors are 6-8%. For tail-latency-sensitive SLO decisions, this error magnitude could lead to incorrect policy choices near decision boundaries. Have you characterized the failure modes, i.e., cases where what-if predictions led to the wrong policy ranking?

**Limitations:**

The authors discuss several limitations in Section 5.9 and Appendix F.3, including coordination overhead for fine-grained operators, centralized backend bottlenecks, and reliance on user-specified contracts. These are honest and relevant. However, two important limitations are underaddressed: (1) the evaluation does not test with production-scale models or multi-GPU inference backends, which significantly limits the generalizability of the results, and (2) the paper does not discuss the overhead of SAGE's runtime itself (e.g., memory footprint of the execution graph, compilation time, overhead of per-replica metrics collection at scale). The societal impact discussion is thorough and appropriate.

**Strengths And Weaknesses:**

Strengths:

- Well-motivated problem: The observation that LLM inference is increasingly a multi-stage pipeline problem, not just a single-model serving problem, is timely and important. The paper clearly articulates why RPC-based composition obscures cross-stage interference and limits whole-pipeline optimization.

- Clean abstraction design: The operator contract model (resource, state, I/O declarations) and the compilation from logical to physical execution graphs is a principled approach. Theorem 3.1 on semantic preservation under parallelization is a useful formal contribution that gives developers confidence in correctness during optimization.

- Controlled experimental methodology: The approach of swapping scheduling policies while holding the compiled execution graph fixed is a strong experimental design choice. It cleanly isolates the effect of policy from graph structure, and Table 1 demonstrates meaningful trade-offs across five policies.

- Comprehensive appendix material: The paper includes extensive supplementary content covering queueing-theoretic analysis (Appendix H), what-if analysis mechanisms (Appendix G), attribution validation (Appendix I), and full reproducibility configs. This level of detail is commendable.

- Practical system design: The five-layer architecture with strict downward dependencies, the pluggable backend integration via OpenAI-compatible endpoints, and the declarative DSL (Listing 1) suggest a system that could be practically adopted.

Weaknesses:

- Limited and somewhat artificial evaluation setup: The primary evaluation uses a 16-node CPU cluster with a single A6000 GPU running Qwen2.5-3B-Instruct. This is a very modest setup relative to the claims about production inference pipelines. Real production LLM deployments involve multi-GPU nodes, larger models, and substantially different bottleneck profiles. The 3B parameter model does not stress GPU memory management, KV-cache pressure, or prefill/decode disaggregation, which are central challenges in production. It is unclear whether SAGE's benefits hold at realistic production scale.

- Lack of baselines against existing systems: The paper does not compare SAGE against any existing pipeline orchestration system (e.g., Ray Serve, LangChain, LlamaIndex, or even a hand-tuned RPC pipeline). All comparisons are internal (SAGE with different policies). Without external baselines, it is difficult to assess whether the improvements are due to the novel abstractions or simply due to reasonable engineering of a new system. The 57% P99 reduction is against simultaneous admission within SAGE itself, not against an alternative system.

- Overclaimed novelty and contributions: The paper repeatedly claims to be "the first" pipeline-as-compilation-target abstraction, but dataflow compilation for distributed execution is well-established (Naiad, Flink, Spark, Differential Dataflow). The novelty is in applying these ideas to LLM inference pipelines specifically, which is worthwhile but should be stated more precisely. The distinction from, say, compiling a Flink DAG with operator-level resource contracts is not made sufficiently clear.

- Theorem 3.1 is somewhat trivial: The semantic preservation theorem states that correct replication with appropriate routing preserves dataflow semantics. This is a standard property of dataflow systems and the proof sketch does not address the hard cases (e.g., non-deterministic operators, partial failures during state migration). The theorem's conditions are strong enough that the result is largely definitional.

- Queueing-theoretic analysis relies on strong assumptions: The M/M/1 analysis in Appendix H assumes Poisson arrivals and exponential service times. The authors acknowledge that LLM inference exhibits heavy-tailed latency distributions, and indeed report 6-8% underestimation of tail behavior. For a systems paper claiming to provide tail-latency control, a more realistic queueing model (e.g., M/G/1 with Kingman's bound) would strengthen the analysis.

- What-if analysis and attribution are evaluated on synthetic/controlled scenarios: The injected bottleneck experiments (Table 5) and the user study (8 participants, small scale) provide limited evidence for real-world applicability. The 87.5% precision figure comes from only 8 flagged operators, making the confidence interval quite wide.

- Writing quality is generally good but suffers from excessive self-promotion: Phrases like "paradigm shift," "fundamentally solves," and "architecturally unique" appear frequently and undermine the paper's credibility. The novelty takeaway boxes, while intended to help, often overstate the contribution relative to prior dataflow systems work.

- The paper defers too much to the appendix: Key design details (compilation rules, flow control bounds, what-if mechanisms, attribution pipeline) are all in appendices. The main text reads more like an extended abstract with pointers. A reader relying only on the main 8 pages would struggle to evaluate the technical depth of several claimed contributions.

---

> ### Author Rebuttal · Authors · 2026-03-31
>
> ## Response: On the evaluation scale and the use of a 3B model
>
> *[Addresses: Weakness 1; Key Question 2]*
>
> We agree this concern is valid. The current paper does not provide 70B+ multi-GPU evidence, and we will state this limitation explicitly.
>
> Our scope is pipeline-level orchestration above the model engine. Tensor parallelism, KV-cache policies, and prefill/decode disaggregation are backend concerns; SAGE manages cross-stage flow control, admission, and scheduling. The 3B setup was chosen to isolate these orchestration effects. In revision, we will make this scope boundary explicit and avoid implying that large-model performance has been empirically established here.
>
> ---
>
> ## Response: On external baselines
>
> *[Addresses: Weakness 2; Key Question 1]*
>
> We agree. We added direct baselines on a matched RAG setup (same vLLM backend, FAISS index, query set, and `max_tokens=20`) against LangChain v0.3, LlamaIndex v0.11, and Ray Serve v2.40.
>
> Summary: SAGE reaches 18.0 req/s at concurrency 8 versus 9.07 (LlamaIndex), 6.76 (LangChain), and 2.51 (Ray Serve). Under 4-pipeline contention, SAGE retains 87% throughput (11.0/12.7), versus 85% (LlamaIndex) and 29% (Ray Serve). We will include full tables and methodology in the revision. This uses the same baseline protocol reported consistently across our rebuttal responses.
>
> ---
>
> ## Response: On novelty claims and writing tone
>
> *[Addresses: Weaknesses 3, 7]*
>
> We agree and will substantially soften language (removing phrases such as "paradigm shift" and "fundamentally solves") and delete overstated novelty boxes.
>
> Our intended claim is narrower: adapting dataflow compilation to multi-stage LLM serving with LLM-specific runtime mechanisms (contracted operators, bounded-queue tail control, policy swap on fixed graphs, what-if replay, and per-operator attribution). We will add clearer positioning against prior dataflow systems and include Apache Flink Agents as concurrent related work.
>
> ---
>
> ## Response: On contract misspecification sensitivity
>
> *[Addresses: Key Question 3]*
>
> We agree this is important and currently under-evaluated. We have not yet reported a full degradation curve for 2x/5x misspecification, and we will state this limitation explicitly.
>
> Current behavior is: categorical contract errors are usually fail-fast and visible; continuous errors mostly degrade placement quality. Existing mitigations include profiled default contracts for common operators, attribution traces, and what-if replay for iterative correction. We will add practical guidance (defaults, tuning workflow, and diagnostics) and frame automatic contract calibration/inference as future work.
>
> ---
>
> ## Response: On non-deterministic operators and Theorem 3.1
>
> *[Addresses: Weakness 4; Key Question 4]*
>
> We agree Theorem 3.1 is primarily a specification result, not a deep new theorem. Its role is to justify the fixed-graph policy comparison protocol.
>
> For non-deterministic operators, our practical guarantees are limited: item preservation and partition-consistency still hold, while order guarantees apply only to deterministic operators. We will make this boundary more explicit and add diagnostics guidance for latency-variance-heavy operators; this will be documented together with the control-flow guarantee boundaries in one appendix section.
>
> ---
>
> ## Response: On what-if analysis accuracy and failure modes
>
> *[Addresses: Weakness 6; Key Question 5]*
>
> Great point. We now describe the main failure mode explicitly: P99 is directionally underestimated, so ranking can be unreliable when policy gaps are small.
>
> Empirically, what-if remains useful when performance gaps are not near-tied (for example FIFO to LoadAware: predicted 19% gain vs. actual 24%). We will add a decision-safety note in the paper: treat near-boundary recommendations as low confidence and confirm with short online trials.
>
> For attribution, we will keep both controlled validation and production-trace evidence, but present the small-sample caveat more explicitly.
>
> ---
>
> ## Response: On appendix dependence and the M/M/1 assumption
>
> *[Addresses: Weaknesses 5, 8]*
>
> We agree on both points.
>
> First, the main text currently offloads too much detail; we will move the core compilation rule summary, queue-bound design rationale, and what-if procedure into the main body.
>
> Second, M/M/1 is used as a first-order planning model, not as a precise tail predictor. The runtime control mechanism (bounded queues + admission) is independent of Poisson/exponential assumptions. We will clarify this and add M/G/1-style tail-aware analysis as a concrete extension direction.

---

> > ### Author Rebuttal · Reviewer_tHWN · 2026-04-03
> >
> > ## Fully Addressed
> >
> > - **External baselines (W2, KQ1):** Added LangChain, LlamaIndex, and Ray Serve comparisons on a matched RAG setup. SAGE reaches 18.0 req/s vs. 9.07 (LlamaIndex), 6.76 (LangChain), 2.51 (Ray Serve). This was the biggest gap and is now closed.
> > - **Novelty overclaims and tone (W3, W7):** Agreed to remove "paradigm shift," "fundamentally solves," overstated novelty boxes, and to add clearer positioning against prior dataflow systems (including Flink Agents).
> > - **Appendix dependence and M/M/1 (W5, W8):** Will move core compilation rules and what-if procedure into the main body. Clarified M/M/1 is a planning model, not a tail predictor; M/G/1 analysis deferred to future work.
> > - **Theorem 3.1 and non-determinism (W4, KQ4):** Acknowledged the theorem is a specification result. Clarified guarantees for non-deterministic operators are limited to item preservation and partition-consistency.
> >
> > ## Partially Addressed
> >
> > - **Evaluation scale / 3B model (W1, KQ2):** Conceded the concern is valid but provided no new large-scale evidence. Argued SAGE targets orchestration above the model engine. The empirical question of whether pipeline optimizations matter when generation dominates remains open.
> > - **Contract misspecification (KQ3):** Acknowledged as under-evaluated. Described existing mitigations (profiled defaults, attribution traces) but did not produce the requested 2x/5x degradation curve. Framed auto-calibration as future work.
> > - **What-if failure modes (W6, KQ5):** Identified the main failure mode (P99 underestimation near decision boundaries) and will add a safety note, but provided only one example rather than a systematic analysis of ranking failures.
> >
> > ## Not Addressed
> >
> > - Partial failure handling in Theorem 3.1's proof sketch.
> > - Small sample size for attribution precision (8 flagged operators, wide confidence interval).

---

> > > ### Author Response · Authors · 2026-04-06
> > >
> > > Thank you for your reconsideration. We would like to clarify that SAGE goes beyond an LLM inference engine such as vLLM or SGLang. It supports scale-out, backpressure management, cross-stage coordination, resource utilization optimization, fault-tolerant serving, distributed execution, and the management of complex serving pipelines above the model engine. These capabilities are especially important in multi-stage serving scenarios that are not directly handled by an inference engine alone, such as retrieval-augmented pipelines, cascaded model stages, tool-use workflows, and other applications with cross-stage dependencies, distributed systems requirements, and system-level orchestration needs.
> > >
> > > With this scope in mind, we respond to the remaining concerns as follows:
> > >
> > > 1) **Larger-model settings.** We will qualify stronger performance extrapolation.
> > >
> > > 2) **Contract misspecification.** We will limit the claim to effectiveness under reasonably specified contracts.
> > >
> > > 3) **What-if analysis.** We will position it as a directional planning tool rather than a high-confidence ranking oracle.
> > >
> > > 4) **Theorem 3.1.** We will clarify that it is a specification-preservation result rather than a fault-tolerance theorem.
> > >
> > > 5) **Attribution precision.** We will moderate the claim strength given the current sample size.

---

### Official Review · Reviewer_Fep9 · 2026-03-12

**Soundness:** 3
**Presentation:** 4
**Significance:** 3
**Originality:** 4
**Overall Recommendation:** 5
**Confidence:** 3

**Summary:**

The article presents SAGE, a framework which treats LLM inference pipelines as dataflows to optimize their scheduling, their placement and the resource sharing. All pipelines need to define contracts which consist in the quantity of requested resources (CPU, GPU and memory), the state of the model and the I/O behavior of the inference. The article claims gaining an 11x speedup factor for the RAG steps at 16 nodes with a dedicated GPU server for LLM.

**Compliance With Llm Reviewing Policy:**

Affirmed.

**Final Justification:**

The rebuttal addressed my main concerns and reinforced my prior recommendation.

**Key Questions For Authors:**

1. Some clarifications can be added to the main paper to indicate what the “compute” workload consists of? We suppose the “mixed” one is a mix between “compute” and “rag”, does it mean some inferences of the workload do not ask for RAG computation?

2. In the evaluation part, table 1 reports some metrics about the scheduler performance for the RAG workload. But the relations between the metrics are not clear enough. For instance, FIFO seems to present relatively good SLOs, considering the throughput and the average latency, but the load balancing is really low compared to the other algorithms. And it does not seem that the 0.5s difference of the p99 is a bad compromise either. Still, the writing implies the algorithm is not good according to the stability. The question is how the poor load balancing does not impact the other performance metrics? Is it due to an eventual overhead of the load-aware strategies?

2. For now, the contracts are user-specified, and as stated by the limitations section of the article, in case the specifications are inaccurate, the system will underperform. Are there hints to help users specify the contracts of their applications? Or maybe include a confidence score to the contract? As predicting the resources requested by user applications is complicated if not impossible, we do not see other ways of counting on what the user gives to the system.

**Limitations:**

yes

**Strengths And Weaknesses:**

The article is easy-to-read and well-constructed. The overview of the system and its architecture are sufficiently described and allow its correct understanding. Multiple points of design are detailed and served as points of interest for the evaluation of the framework. The evaluation experiments target a sufficient set of workloads to estimate the benefits of SAGE.

The main drawback of the approach is clearly identified in the article, which is the need for users to accurately estimate the resources needed by their applications, and is recognized as a limitation in the final part of the article.

Some clarifications may be added to the evaluation part to help understand the results of the experiments better.

---

> ### Author Rebuttal · Authors · 2026-03-31
>
> ## Response: On the COMPUTE and MIXED workloads
>
> *[Addresses: Key Question 1]*
>
> We agree and will make both workloads explicit in the main paper.
>
> **COMPUTE** is a synthetic CPU-bound pipeline (`Source -> ComputeOp -> Sink`) used to isolate orchestration overhead. `ComputeOp` runs a deterministic busy loop (~3.5 ms/operator call). End-to-end latency (~26 ms) is dominated by coordination cost (dispatch/serialization/queue transit), which explains its lower scale-out (3.8x at 16 nodes) versus RAG (11.4x), where longer model service time amortizes per-hop overhead.
>
> **MIXED** is a strict 50/50 interleaving of COMPUTE and RAG requests in the same cluster. So yes: half of requests do not trigger RAG. Its role is to create a bimodal service-time distribution and expose head-of-line blocking effects under simple queueing policies. We will add these definitions and generation rules to Section 5.1.
>
> ---
>
> ## Response: On the relation among throughput, latency, P99, and load balance in Table 1
>
> *[Addresses: Key Question 2]*
>
> Great question. The short answer is:
>
> 1. At the Table 1 operating point (moderate load), FIFO's imbalance mainly hurts tails, not mean latency/throughput, because hotspots are not yet in saturation.
> 2. Yes, load-aware policies add dispatch overhead and can reduce batching opportunities, so better balance does not automatically maximize throughput.
>
> We agree our original wording over-penalized FIFO. We will revise Section 5.3 to present explicit trade-offs: FIFO has strong throughput (18.5 req/s) and competitive mean latency (2.5 s), but weaker balance (52%) and slightly worse P99 (+0.5 s). Load-aware policies improve tail stability and robustness under rising load, at some throughput cost.
>
> To make this relationship clearer, we will add per-node utilization and queue-depth breakdowns alongside Table 1.
>
> ---
>
> ## Response: On user-specified contracts and practical usability
>
> *[Addresses: Key Question 3; Weakness 1]*
>
> We agree this is the main practical risk and appreciate the suggestion.
>
> Our current approach is: contracts are explicit user declarations, but users are not expected to start from scratch. Common operators ship with default contracts from profiling; users usually override only task-specific deltas.
>
> We distinguish two error types. Categorical errors (for example wrong GPU affinity/state semantics) are usually caught early. Continuous errors (CPU/memory/I/O estimates) typically degrade placement quality rather than break functionality. To support iterative correction, SAGE already exposes per-operator attribution traces and what-if replay to compare declared versus observed behavior. We have not yet reported a full 2x/5x misspecification degradation curve and will state this limitation explicitly.
>
> In the revision, we will add practical guidance for contract authoring (recommended starting templates + tuning workflow). We will also discuss the reviewer's confidence-score direction as future work: warm-up calibration, divergence alerts, and interval-valued contract fields. For what-if usage, we will add a decision-safety note that near-boundary policy recommendations should be treated as low confidence and verified with short online trials.

---

> > ### Author Rebuttal · Reviewer_Fep9 · 2026-04-02
> >
> > Thanks for your response. You answered my concerns.

---

### Decision · Program_Chairs · 2026-04-30

**Decision:**

Accept (regular)

**Comment:**

This paper presents SAGE, a framework that treats multi-stage LLM inference pipelines as first-class compilation targets, using declarative operator contracts to enable principled scheduling, replica placement, backpressure control, and explainable performance attribution. The problem is well-motivated and practically relevant, and the paper is generally well-written.

Reviewer opinions are divided with an active rebuttal line:
1. The most critical gap — lack of external baselines — was addressed in the rebuttal with comparisons against LangChain, LlamaIndex, and Ray Serve, showing meaningful throughput advantages (2x over LlamaIndex, 2.7x over LangChain).
2. The overclaimed novelty language was also agreed to be removed.
3. Several concerns remain partially unresolved: the evaluation uses only a 3B parameter model on a single GPU, leaving open whether pipeline-level optimizations matter at production scale; the contract misspecification degradation curve was not provided.

Due to the limitation of the computational resource (GPUs), testing 3B models is fine However, the authors are strongly encouraged to narrow the scope claims to match the demonstrated setting, provide the misspecification sensitivity analysis, and incorporate all rebuttal results into the final version.